# WORKFLOWLLM: ENHANCING WORKFLOW ORCHESTRATION CAPABILITY OF LARGE LANGUAGE MODELS

**Shengda Fan**[1*], **Xin Cong**[2*†], **Yuepeng Fu**[2], **Zhong Zhang**[2], **Shuyan Zhang**[3], **Yuanwei Liu**[4], **Yesai Wu**[2], **Yankai Lin**[1†], **Zhiyuan Liu**[2], **Maosong Sun**[2]

[1]Renmin University of China [2]Tsinghua University [3]The University of Manchester
[4]Wuhan University
fanshengda,yankailin@ruc.edu.cn, congxin1995@mail.tsinghua.edu.cn

## ABSTRACT

Recent advancements in large language models (LLMs) have driven a revolutionary paradigm shift in process automation from Robotic Process Automation to Agentic Process Automation by automating the workflow orchestration procedure based on LLMs. However, existing LLMs (even the advanced OpenAI GPT-4o) are confined to achieving satisfactory capability in workflow orchestration. To address this limitation, we present WorkflowLLM, a data-centric framework elaborately designed to enhance the capability of LLMs in workflow orchestration. It first constructs a large-scale fine-tuning dataset WorkflowBench with $106,763$ samples, covering $1,503$ APIs from 83 applications across 28 categories. Specifically, the construction process can be divided into three phases: (1) Data Collection: we collect real-world workflow data from Apple Shortcuts and RoutineHub, transcribing them into Python-style code. We further equip them with generated hierarchical thought via GPT-4o-mini. (2) Query Expansion: we prompt GPT-4o-mini to generate more task queries to enrich the diversity and complexity of workflows. (3) Workflow Generation: we leverage an annotator model trained on collected data to generate workflows for synthesized queries. Finally, we merge the synthetic samples that pass quality confirmation with the collected samples to obtain the WorkflowBench. Based on WorkflowBench, we fine-tune Llama-3.1-8B to obtain WorkflowLlama. Our experiments show that WorkflowLlama demonstrates a strong capacity to orchestrate complex workflows, while also achieving notable generalization performance on previously unseen APIs. Additionally, WorkflowBench exhibits robust zero-shot generalization capabilities on an out-of-distribution task planning dataset, T-Eval. Our data and code are available at https://github.com/OpenBMB/WorkflowLLM.

## 1 INTRODUCTION

Process Automation (PA) (Cichocki et al., 1997), as a long-standing pursuit of the human race, aims to automate repetitive tasks to minimize human labor and improve efficiency. Robotic Process Automation (RPA), the current predominant PA technique, abstracts the repetitive task into a workflow (i.e., a program that can execute automatically) by orchestrating various actions (e.g., functions or APIs) (Ivančić et al., 2019; Hofmann et al., 2020; Wewerka & Reichert, 2020; Agostinelli et al., 2020; Ferreira et al., 2020). While RPA successfully reduces the human labor via automated workflow execution, the process of orchestrating workflows still requires substantial manual effort. Recently, large language models (LLMs) (OpenAI, 2022; 2023; Touvron et al., 2023a;b; Dubey et al., 2024) have achieved remarkable performance beyond natural language processing (Ahn et al., 2022; Cheng et al., 2023; Qian et al., 2024). The emergence of LLMs has unveiled a paradigm shift trend, moving from Robotic Process Automation to Agentic Process Automation (APA) (Ye et al., 2023; Zeng et al., 2023; Huang et al., 2024; Wornow et al., 2024; Li et al., 2024) which automates the workflow orchestration process by utilizing LLMs to build the workflow.

---

[*] Indicates equal contribution.
[†] Corresponding author.

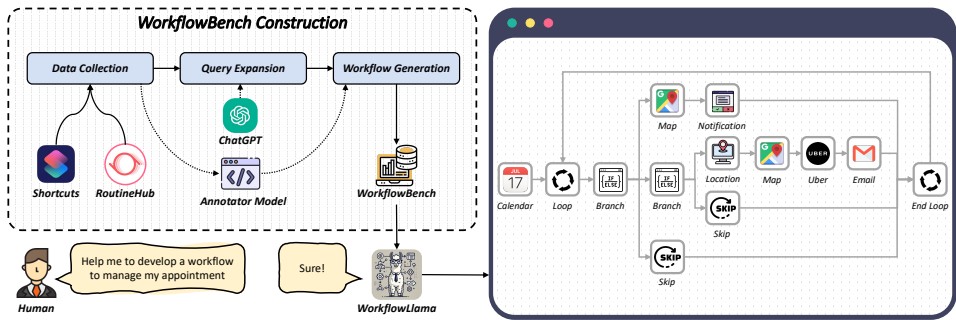

Figure 1: Overview of WorkflowLLM. It first constructs WorkflowBench through a three-phase pipeline and fine-tunes WorkflowLlama, which can generate workflows based on the user's query (appointment management in this case).

The paradigm shift is hindered by **LLMs' limited ability to orchestrate complex workflows**, leading to two main limitations in current APA methods: (1) **Limited Action Scale**: LLMs, even advanced models like GPT-4, can only orchestrate workflows with an average of 6.1 actions, falling short of real-world complexity requirements (Ye et al., 2023). In contrast, Apple Shortcuts typically involves 70.4 actions. (2) **Simple Logical Structure**: Most current methods focus on sequential actions (Yao et al., 2022; Qin et al., 2024; Chen et al., 2024), while real-world workflows require more complex structures, including nested branches/loops, as seen in Apple Shortcuts' average of 2.6 nested logical structures. Thus, **there is an urgent need to unlock the workflow orchestration capability of LLMs to expedite the paradigm shift in process automation.**

To address these challenges, we propose **WorkflowLLM**, a data-centric framework including dataset construction, model training, and evaluation to enhance LLMs' workflow orchestration capabilities (shown in Figure 1). Specifically, we first construct WorkflowBench, the first dataset explicitly designed for workflow generation. WorkflowBench consists of $106,763$ instances, encompassing $1,503$ APIs across $83$ applications, structured through three primary phases:

- **Data Collection**: We select shortcuts from RoutineHub as high-quality data sources because they represent a robust RPA application with numerous expert-developed workflows available. We curate $14,771$ human-annotated, high-quality shortcuts spanning $28$ diverse categories, alongside associated metadata including titles, functionality descriptions, and API documentations. As the raw shortcuts are not suitable for LLMs to process, and considering that Python allows convenient parameter passing and control logic (Wang et al., 2024b), we transcribe the shortcuts into Python code. Subsequently, we prompt GPT-4o-mini to generate comments, task plans, and task queries at varying levels of granularity—from fine-grained to coarse-grained—to enrich the data with detailed thought processes and enhance the learning efficacy of LLMs (Wei et al., 2023).

- **Query Expansion**: To enrich the diversity and complexity of workflows, we utilize GPT-4o-mini to generate additional task queries. Specifically, we first sample applications with diverse functionalities and select their APIs, along with built-in APIs, to prompt GPT-4o-mini to generate task queries that leverage these sampled APIs to accomplish specific tasks. To further ensure workflow complexity, we also sample real-world workflow examples as demonstrations to guide GPT-4o-mini in generating similar workflows.

- **Workflow Generation**: Since existing LLMs, including GPT-4o, struggle with workflow generation, we first train a workflow annotator model on real-world shortcuts. We then use the trained annotator to generate workflows for expanded task queries. To maintain dataset quality, we perform a quality confirmation step. Specifically, GPT-4o-mini refines the workflows to fix minor bugs, followed by rule-based filtering to remove workflows with logical errors.

To evaluate the capability of LLMs in workflow orchestration, we employ two metrics: the reference-code-based metric **CodeBLEU** and the model-based metric **Pass Rate**. Experimental results demonstrate that WorkflowLlama consistently and significantly outperforms all baselines, including GPT-4o even with the in-context learning technique, across both metrics under unseen instructions and unseen APIs settings. Furthermore, WorkflowBench demonstrates strong generalization capabilities in out-of-distribution (OOD) scenarios, particularly on the T-Eval benchmark (Chen et al., 2024), where it achieves an F1 plan score of **77.5**%.

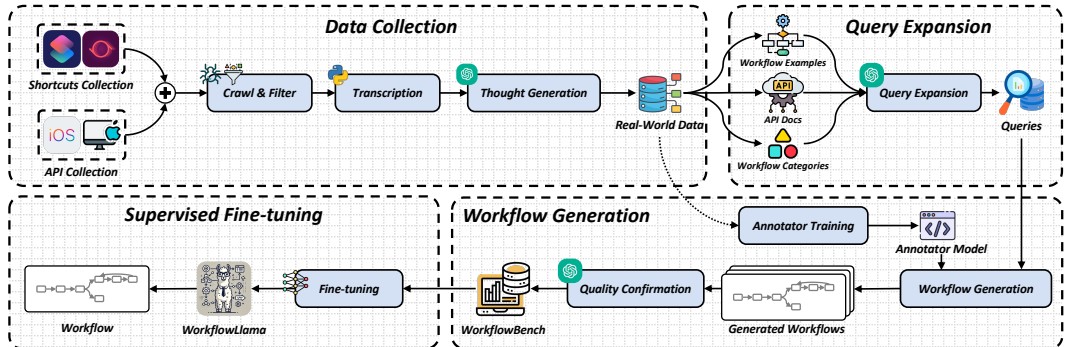

Figure 2: Illustration of our WorkflowLLM which contains three phases to construct Workflow-Bench, followed by the supervised fine-tuning phase to derive WorkflowLlama.

## 2    RELATED WORK

**Process Automation**    RPA has gained considerable attention for automating repetitive tasks in various productivity scenarios (Ivančić et al., 2019; Hofmann et al., 2020; Wewerka & Reichert, 2020; Agostinelli et al., 2020; Ferreira et al., 2020). RPA predominantly relies on handcrafted workflows (e.g., programming, recording human behavior), making them highly suitable for automating well-structured, routine processes (Herm et al., 2020). However, such approaches require substantial efforts and in-depth domain expertise, resulting in high setup costs and limited adaptability. Recent advancements in LLMs have spurred interest in integrating these models into RPA to enhance flexibility and reduce dependency on manual workflow creation. Ye et al. (2023) introduced the concept of APA, which utilizes LLMs to autonomously orchestrate workflows based on human instructions. Subsequently, several studies have sought to apply APA in various domains, including travel planning (Xie et al., 2024), smartphone applications (Huang et al., 2024), enterprise automation (Wornow et al., 2024), financial question answering (Zeng et al., 2023), and data analysis (Li et al., 2024). Despite relying on advanced LLMs (e.g., GPT-4), these approaches have often exhibited suboptimal performance, highlighting challenges faced by existing LLMs in workflow orchestration. While Li et al. (2024) made an effort to fine-tune Mixtral-8×7B (Jiang et al., 2024), it could only orchestrate sequential workflows with an average of 15.6 actions, remaining insufficient for real-world requirements. This work addresses a critical gap by proposing WorkflowLLM framework to enhance the workflow orchestration capabilities of LLMs to meet real-world demands.

**Tool Learning**    Workflow orchestration driven by LLMs frequently depends on external tools to extend their operational capabilities. Recent studies show that LLMs can acquire and utilize tools by learning from documentation, enabling them to tackle complex tasks beyond their native abilities (Wu et al., 2023; Schick et al., 2024; Qin et al., 2023b; 2024). This integration allows LLMs to access real-time knowledge and perform specialized operations, particularly for intricate processes (Yang et al., 2023; Nakano et al., 2021; Qin et al., 2023a; Wang et al., 2024c; Gao et al., 2023). To further enhance this capability, several efforts have introduced datasets to fine-tune LLMs for tool interaction (Zhuang et al., 2024; Qin et al., 2024; Wang et al., 2024a). However, they have a limited action scope and lack complex logic control, focusing primarily on dynamically selecting APIs and filling parameters. Compared to tool learning scenarios, orchestrating workflows demands more sophisticated planning and reasoning that current LLMs have yet to fully realize. In response to these limitations, we present WorkflowLLM to significantly improve LLMs' capabilities in workflow orchestration. Besides, Shen et al. (2024) also used Apple's Shortcuts but aimed to assess LLMs' tool utilization ability. In contrast, we emphasize a different scenario, workflow orchestration and aim to enhance the workflow orchestration ability rather than evaluation alone.

## 3    WORKFLOWLLM

As Figure 2 shows, WorkflowLLM introduces a data-centric framework to enhance the capability of LLMs in workflow orchestration by constructing a high-quality supervised fine-tuning dataset

WorkflowBench. In this section, we outline the dataset construction process, which is carried out in three distinct phases: Data Collection, Query Expansion, and Workflow Generation.

## 3.1 DATA COLLECTION

We first introduce Apple Shortcuts and RoutineHub, and describe how we crawl and filter to get high-quality data. We then convert the shortcuts into Python-style workflow code. Finally, we prompt GPT-4o-mini [1] to generate hierarchical thoughts for each shortcut.

**Apple Shortcuts and RoutineHub**    Apple Shortcuts, as a representative application of RPA, is developed by Apple Inc. This tool facilitates the automation of a series of actions, enabling users to efficiently perform a diverse range of tasks. The actions within Shortcuts are APIs provided by both built-in Apple applications, such as *Safari*, and third-party applications like *OpenAI*. Each application may provide multiple actions. For instance, *OpenAI* provides APIs that facilitate voice conversations and text interactions with ChatGPT. Through a simple drag-and-drop interface, users can construct complex workflows, such as navigating to the nearest coffee shop or downloading watermark-free images from TikTok.

RoutineHub[2] is a prominent community for sharing shortcuts, with a collection of thousands of shortcuts across both iOS and macOS platforms. All shortcuts on RoutineHub are categorized into 28 workflow categories (e.g., Business, Health & Fitness, Productivity, etc). RoutineHub records the metadata of each shortcut (e.g., title, description, iCloud URL), providing valuable information.

**Crawling and Filtering**    For each shortcut, we crawl the title, developer-provided description, and iCloud URL linked to Apple. As RoutineHub does not provide the source code for these shortcuts, we further crawl it from their iCloud URLs. Besides, we merge shortcuts collected by Shortcuts-Bench (Shen et al., 2024), sourced from platforms like ShareShortcuts[3] and MacStories[4], to further expand the scale of our dataset. However, the source code of these shortcuts lacks detailed information about the involved actions, such as API metadata. Inspired by ShortcutsBench (Shen et al., 2024), we extract action information from macOS's built-in definition files and third-party application interface definition files. For each API, we record its name, description, parameter names, parameter types, default values, return value types, and return value name, which provides a valuable resource for LLMs to efficiently interpret and utilize these APIs, even in zero-shot scenarios.

To ensure compatibility between the crawled shortcuts and the action interfaces, we implement a stringent filtering mechanism to verify that all API calls are executed correctly. During this process, we identify that some shortcuts contain non-interpretable binary sequences as API parameters, potentially disrupting the training process of language models. To maintain data quality, we remove these samples from the dataset. As a result, we curate a final set of $14,771$ high-quality shortcuts, ensuring the reliability of the dataset for subsequent data expansion and model training.

**Shortcuts Transcription**    The original shortcut source codes are written in property lists format (Hummert & Humphries, 2022), which sequentially encodes logical constructs like branches and loops. This encoding is notably different from the types of data commonly used in the pre-training of LLMs. To address this gap, we convert the shortcuts into abstract syntax trees (ASTs), apply pre-order traversal to transform them into Python code, with further algorithmic details provided in Appendix A. Furthermore, the original shortcuts use hexadecimal strings as variable names, leading to reduced semantic clarity. To improve interpretability, we use GPT-4o-mini to automatically reassign these variables with more contextually meaningful names, thereby enhancing the overall readability and utility of the code for further language model training. A typical comparison between property lists and Python code can be found in Appendix E.

**Thought Generation**    To provide informative guidance for LLMs in orchestrating workflows, we design a three-level thought hierarchy from fine-grained to coarse-grained: (1) **Low-level comments**

---

[1]We selected gpt-4o-mini-2024-07-18 due to its balanced performance and cost efficiency.
[2]https://routinehub.co/
[3]https://shareshortcuts.com
[4]https://www.macstories.net/shortcuts

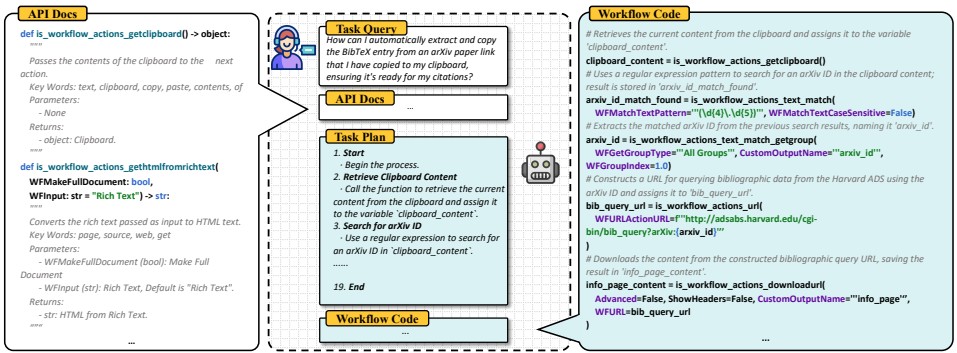

Figure 3: Illustration of data field composition in WorkflowBench comprising *Task Query*, *API documentations*, *Task Plan*, and *Workflow code* with *Comments*.

are intended to clarify the purpose of each action within the workflow. (2) **Median-level plans** represent an abstraction over a sequence of actions, outlining the collective goal of these steps. (3) **High-level queries** reflect the user's requirements, specifying the intended outcome without prescribing specific methods to achieve it. These three levels of thought are generated through a bottom-up approach. Specifically, given the transcribed workflow $w$, let the set of actions within the workflow be denoted as $\mathcal{A}$, where each action $a_i \in \mathcal{A}$ corresponds to a Python function call. The documentation $d_i \in \mathcal{D}$ associated with each action includes key information such as the function name, input and output parameters, and a description of its functionality. For each action $a_i$, we generate a corresponding comment $c_i$ by prompting GPT-4o-mini. Subsequently, given the action set $\mathcal{A} = \{a_i\}$ and comments $\mathcal{C} = \{c_i\}$ of workflow $w$, we prompt GPT-4o-mini to generate the corresponding task plan $\mathcal{P}$. We combine the task plan $\mathcal{P}$, the comments $\mathcal{C}$, and the action set $\mathcal{A}$ of the workflow $w$ to generate the high-level task query $\mathcal{Q}$. This bottom-up approach, resembling the summarization task, may contribute to content reliability and potentially reduce the risk of hallucination.

Finally, as Figure 3 shows, each workflow $w$ is represented as: $w = \{\mathcal{Q}, \mathcal{D}, \mathcal{P}, \mathcal{A}\}$, where the workflow $w$ consists of the task query $\mathcal{Q}$, action documentation $\mathcal{D}$ for all involved actions, the task plan $\mathcal{P}$, and all actions represented as annotated Python code $\mathcal{A}$. A detailed example can be found in Appendix F.

## 3.2 QUERY EXPANSION

After performing a comprehensive statistical analysis on the collected data, we find that the data exhibits significant complexity, with an average of 70.4 actions and 12 branches, surpassing the complexity of existing workflow-related benchmarks. However, the diversity of the data is relatively low. Specifically, 40.3% of the workflows fall under the `Utilities` category, and over 99% of the APIs used are Apple's built-in APIs (i.e., those classified as `is_workflow_actions` APP).

Therefore, we intend to expand the dataset by focusing on two key aspects: (1) **Diversity**: making up for the lack of diversity in real data and covering a broad range of APIs and workflow categories to enhance the model's utility and robustness; (2) **Complexity**: matching the action scale and logical complexity of the real-world data to ensure that they can effectively represent real-world problems and orchestrate nodes accordingly. To this end, we sample APIs from diverse applications and multiple workflows with representative logical structures (e.g., whether they contain branches or loops) to synthesize additional queries.

To ensure that the number of APIs in the synthesized dataset aligns with real-world usage, we sample $n$ APIs based on real-world distributions. Approximately $\lfloor n/2 \rfloor$ are drawn from Apple's built-in API set (e.g., *openurl* or *sendemail*), with the remainder from third-party applications (e.g., *OpenAI*). The total number of built-in and external APIs is thus $n$.

To ensure that the sampled APIs can interact coherently, we do not sample directly from the entire API set. Instead, we first randomly select 1-5 applications and then choose all APIs from these selected applications. This method ensures that the selected APIs are functionally compatible and capable of representing real-world workflows.

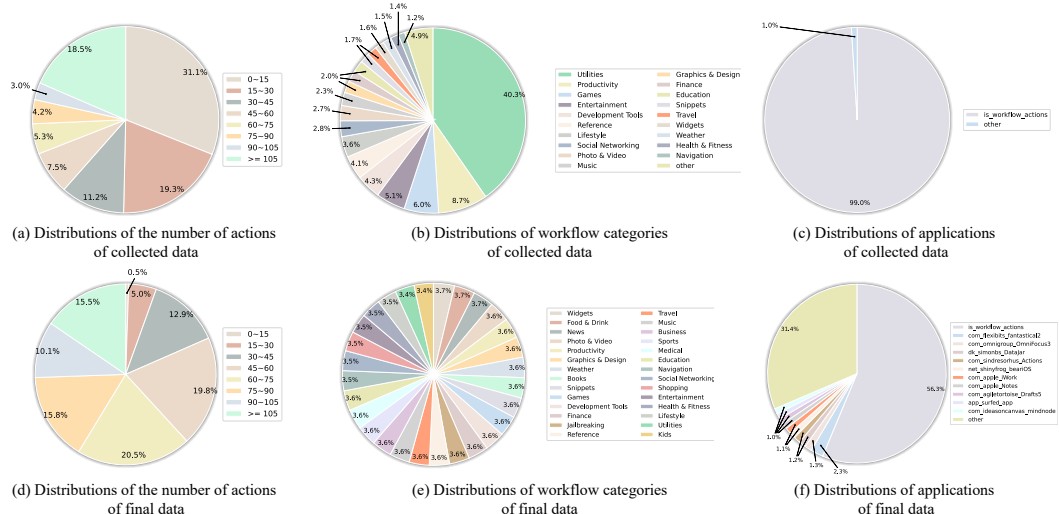

Figure 4: Comparison of the distributions across workflow categories, APPs, and action counts in the collected data and the final dataset. The upper section shows the original data collected from Apple Shortcuts and RoutineHub, while the lower section presents the expanded dataset distributions.

The prompt used for GPT-4o-mini to synthesize queries consists of four components: (1) a general prompt to describe the task query generation task, (2) documentations for the sampled APIs, (3) in-context examples from the collected data for reference, and (4) the workflow category to which the query belongs to. By controlling the workflow category and in-context examples, we can ensure the diversity and complexity of the generated data. As seen from Figure 4, the synthesized query has a more balanced category distribution and uses more third-party APIs. Although most of the APIs used are still built-in APIs, this is reasonable considering that they carry necessary operations.

## 3.3 WORKFLOW GENERATION

To annotate the corresponding workflows of the synthesized queries effectively, we train an annotator model based on the collected shortcuts data to support more diverse applications and categories, while ensuring consistency with the real-world data as much as possible.

**Annotator Training**   First, we construct the supervised fine-tuning (SFT) dataset based on the collected human-labeled shortcuts. Specifically, each workflow data point comprises a query $\mathcal{Q}$, the corresponding action documentation $\mathcal{D}$, the task plan $\mathcal{P}$, and the workflow represented as annotated Python code $\mathcal{A}_{commented}$. During the SFT process, as shown in Figure 3, we take the query $\mathcal{Q}$, the corresponding action documentation $\mathcal{D}$ as the input to guide the

Table 1: Detailed statistics of WorkflowBench. *Seed.* refers to the collected data from Shortcuts. *Train.* and *Test.* refers to the training set and the test set of WorkflowBench respectively.

| Statistics | Seed. | Train. | Test. |
|---|---|---|---|
| Num. of Instances | 14,771 | 105,573 | 1,190 |
| Num. of APPs | 71 | 83 | 31 |
| Num. of APIs | 584 | 1,503 | 324 |
| Num. of Categories | 28 | 28 | 28 |
| Avg. Action | 70.4 | 78.5 | 41.7 |
| Avg. IF | 12.0 | 7.4 | 7.9 |
| Avg. LOOP | 0.7 | 0.5 | 0.5 |
| Avg. Nested Depth | 2.6 | 2.7 | 2.1 |

model to generate a task plan $\mathcal{P}$, followed by the step-by-step generation of the current thought (i.e., the comment $c_i$) and the corresponding action $a_i$, which includes the action name and its associated parameters. We use the trained annotator to generate workflows $\mathcal{A}'$ from synthesized queries.

**Quality Confirmation**   Due to the limited accuracy of the annotator model, the generated workflows may contain errors to some extent. For example, we manually identify issues in $\mathcal{A}'$ (e.g., extraneous branches not relevant to the query and incorrect function call formats). To enhance the overall quality, we prompt GPT-4o-mini with in-context samples to refine both $\mathcal{A}'_{commented}$ and $\mathcal{P}'$.

Then, we use rule-based filtering to remove workflows with fundamental errors. Specifically, we remove samples that don't incorporate code, don't utilize the given APIs, or violate parameter constraints of APIs. More details of quality control protocol can be found in Appendix C.

Finally, we derive a synthesized dataset of $91,992$ instances, which is combined with the initially collected data to form the final WorkflowBench. It contains $106,763$ instances with $1,503$ APIs across $83$ applications, which are used to train WorkflowLlama. The statistics of WorkflowBench are listed in Table 1 and the distribution comparisons of workflow categories, APPs, and the number of actions between the collected data and final data are demonstrated in Figure 4. From the statistical results, we can see that the synthetic data maintains complexity while expanding diversity.

## 4 EXPERIMENTS

### 4.1 EXPERIMENTAL SETUP

**Training Details**  We fine-tune the annotator and WorkflowLlama on LLaMA-3.1-8B (Dubey et al., 2024) for 3 epochs using the AdamW optimizer (Loshchilov & Hutter, 2019). A linear learning rate scheduler is used with a peak learning rate of $2 \times 10^{-5}$ and a warm-up ratio of $0.1$. Each mini-batch contains 32 examples, and the maximum sequence length is set as $8,192$ tokens.

**Baselines**  To provide a comprehensive comparison, we select several representative LLMs as baselines for our experiments. These baselines include proprietary models such as GPT-4o-mini and GPT-4o, as well as open-source models like Qwen2-7B (qwe, 2024), Llama-3.1-8B, and Llama-3.1-70B (Dubey et al., 2024). Additionally, we apply in-context learning (ICL) (Dong et al., 2022) with one random-sampled instance to these baselines to better adapt them for workflow orchestration.

**Metrics**  In the main experiments, we use both reference-code-based metrics and a model-based evaluation to comprehensively evaluate the quality of the generated workflows. For reference-based metrics, we apply **CodeBLEU** (Ren et al., 2020) with four components:

- **BLEU** measures N-gram overlap for token-level similarity.
- **Weighted N-Gram Match** assigns higher weights to critical code tokens like keywords.
- **Syntactic AST Match** compares the Abstract Syntax Trees (ASTs) to assess syntactic accuracy.
- **Semantic Data-Flow Match** evaluates logical correctness by comparing data-flow relationships between variables.

Together, these components provide a comprehensive evaluation of both syntactic and semantic aspects of the workflows. We follow Ren et al. (2020), setting the four components to 0.1, 0.1, 0.4, and 0.4, respectively, and calculate a weighted sum to obtain the CodeBLEU score. For model-based evaluation, we elaborately prompt GPT-4o-mini as the automatic evaluator to evaluate generated workflows' **Pass Rate**, which indicates the proportion of functionally correct workflows generated using only the provided APIs thus is format irrelevant. The evaluation prompt is in Appendix B.2.

### 4.2 EFFECTIVENESS OF EVALUATOR

To validate the reliability of the GPT-4o-mini evaluator in terms of Pass Rate, we sample 30 task queries and workflow code pairs for each model in Table 2, forming a human-evaluated dataset of 330 instances ($30 \times 11 = 330$). First, we use GPT-4o-mini to label whether each instance could complete the given tasks only using the provided APIs. Then, human evaluators re-label the sampled data according to the same criteria specified in Appendix B.2. Ultimately, 268 instances are labeled consistently by both the GPT-4o-mini evaluator and human evaluators, achieving an agreement rate of **81.2**%, demonstrating the reliability and effectiveness of the evaluator.

### 4.3 MAIN EXPERIMENTS

**Settings**  The main experiments are conducted on the test set of WorkflowBench. Ideally, by scaling both the quantity and diversity of instructions and unique tools within the training data, WorkflowLlama is expected to generalize to novel instructions and APIs that are not seen during training. This is particularly important because it enables users to define custom APIs and allows WorkflowLlama to adapt based solely on the provided documentation. To evaluate this capability,

Table 2: Performance comparison of various models on the test set of WorkflowBench under the **unseen instructions (ID)** and **unseen APIs (OOD)** settings (%).

| Model | CodeBLEU | | | | | | | | | | | | Pass Rate | |
|---|---|---|---|---|---|---|---|---|---|---|---|---|---|---|
| | **BLEU** | | **Weighted N-Gram** | | **AST** | | **Data-Flow** | | **Overall** | | | | **Pass Rate** | |
| | ID | OOD | ID | OOD | ID | OOD | ID | OOD | ID | OOD | | | ID | OOD |
| **Proprietary Models** | | | | | | | | | | | | | | |
| GPT-4o-mini | 0.4 | 0.4 | 1.5 | 1.6 | 29.5 | 29.5 | 37.0 | 36.3 | 26.8 | 26.5 | | | 54.8 | 47.5 |
| *w/ ICL* | 0.5 | 0.5 | 1.7 | 1.8 | 35.3 | 34.4 | 35.1 | 34.2 | 28.3 | 27.7 | | | 66.0 | 57.7 |
| GPT-4o | 0.5 | 0.4 | 1.8 | 1.7 | 33.5 | 31.8 | 37.3 | 36.9 | 28.5 | 27.7 | | | 56.6 | 47.5 |
| *w/ ICL* | 0.5 | 0.5 | 1.8 | 1.8 | 37.1 | 35.3 | 38.0 | 36.6 | 30.2 | 30.0 | | | 67.5 | 57.6 |
| **Open-Source Models** | | | | | | | | | | | | | | |
| Qwen2-7B | 0.4 | 0.4 | 1.2 | 1.3 | 27.2 | 27.7 | 33.2 | 33.1 | 24.4 | 24.5 | | | 25.6 | 22.6 |
| *w/ ICL* | 0.5 | 0.5 | 1.2 | 1.3 | 30.2 | 29.8 | 32.4 | 32.9 | 25.2 | 25.3 | | | 28.2 | 26.4 |
| Llama-3.1-8B | 0.6 | 0.7 | 1.2 | 1.4 | 31.0 | 29.6 | 30.0 | 30.8 | 24.6 | 24.3 | | | 33.0 | 24.5 |
| *w/ ICL* | 0.7 | 0.7 | 1.3 | 1.4 | 34.0 | 32.4 | 32.6 | 32.4 | 25.3 | 25.2 | | | 40.2 | 32.7 |
| Llama-3.1-70B | 0.4 | 0.4 | 1.4 | 1.5 | 29.9 | 30.0 | 37.8 | 37.6 | 27.3 | 27.2 | | | 55.4 | 42.3 |
| *w/ ICL* | 0.4 | 0.4 | 1.6 | 1.5 | 34.1 | 32.9 | **39.1** | **38.4** | 29.5 | 28.7 | | | 67.6 | 61.4 |
| WorkflowLlama (8B) | **9.4** | **7.0** | **11.09** | **8.3** | **55.1** | **48.8** | 38.0 | 35.3 | **39.3** | **35.1** | | | **76.9** | **70.4** |

we assess WorkflowLlama's generalization performance at two levels: (1) **Unseen Instructions**, considers an **In-Distribution (ID)** setting, which involves using the same set of APIs as those in the training data, and (2) **Unseen APIs**, considers an **Out-Of-Distribution (OOD)** setting, involving only 50 common APIs required to construct workflows and APIs that are absent from the training data. To minimize API retrieval errors, we use queries and golden APIs as inputs in both ID and OOD settings to assess the workflow orchestration capabilities of different models [5].

**Main Results**   The results are placed in Table 2, from which we derive that:

1. Although multiple workflows can successfully complete a query, there is a positive correlation between the reference-free Pass Rate metric and the reference-based CodeBLEU metric. Given that the Pass Rate metric derived from GPT-4o-mini aligns with human evaluations over 80% of the time, CodeBLEU serves as a reliable proxy for evaluating workflow orchestration capabilities.

2. All models demonstrate a certain capacity for workflow orchestration. This may stem from their inherent instruction-following and code-generation capabilities. We find that models like GPT-4o and Llama-3.1-70B, which perform better on generic tasks, also excel in workflow orchestration. In addition, prompting with in-context samples significantly enhances the models' performance.

3. We find that scores on text overlap metrics such as BLEU and weighted N-gram are low for all models. This is because these metrics only measure the degree of overlap between the candidate and reference texts, and are highly sensitive to implementation details.

4. The workflow orchestrate performance of WorkflowLlama even outperforms powerful closed-source models GPT-4o with ICL by a large margin [6]. Specifically, WorkflowLlama achieves a **39.3**% score on CodeBLEU and a **76.9**% Pass Rate under ID settings, demonstrating the validity of our proposed WorkflowLLM framework and WorkflowBench dataset.

5. WorkflowLlama demonstrates strong generalization capabilities. Even though it has not been trained on the same instructions or APIs, it still significantly outperforms the vanilla Llama-3.1 on all metrics, ahead of or close to the more powerful foundation models. Notably, our method achieves **35.1**% in CodeBLEU and **70.4**% in Pass Rate, outperforming all strong baselines.

### 4.4   ANALYSIS OF WORKFLOW COMPLEXITY

To evaluate the models' ability to generate workflows of varying complexity, we break down the performance of CodeBLEU according to the total number of actions, the number of branches and loops,

---

[5]Additional experimental results and analysis using an LLM retriever are provided in Appendix D.

[6]Additional human-annotated evaluations demonstrate that WorkflowLlama achieves a human pass rate of 71.1%, surpassing GPT-4o with ICL, which attains a human pass rate of 65.9%.

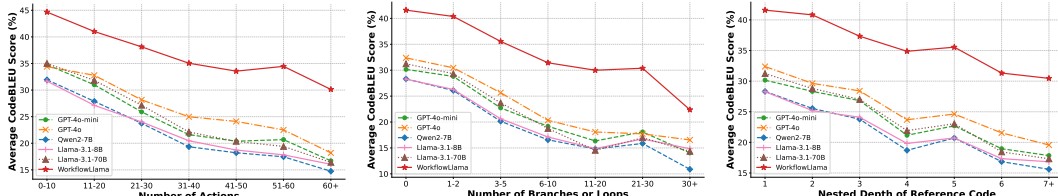

Figure 5: Performance comparisons based on the number of actions, the number of Branch & Loop, and the nested depth of the reference code.

and the nested depth of the reference code. As shown in Figure 5, the performance of all models deteriorates as the number of actions or the logical complexity increases, indicating the challenge of orchestrating complex workflows. However, across all levels of complexity, WorkflowLlama significantly outperforms all other models. Moreover, the relative performance of WorkflowLlama improves as the complexity of the workflow increases, which demonstrates fine-tuning with WorkflowBench significantly enhances the model's ability to handle more complex workflows.

### 4.5 OUT-OF-DISTRIBUTION GENERALIZATION TO T-EVAL (CHEN ET AL., 2024)

**Settings** To further evaluate the generalization capability of WorkflowLlama, we conduct experiments using an OOD benchmark, T-Eval, a widely-used benchmark to evaluate the multi-step decision-making capability of LLMs to utilize APIs. The original data format in T-Eval is based on JSON or strings, which differ significantly from the Python-based format employed in WorkflowBench. To ensure the evaluation metrics' consistency between ours and the original paper, we convert WorkflowBench into JSON format while preserving the metadata of workflows and the specifics of queries. Subsequently, we retrain WorkflowLlama on the transformed dataset. We employ the **F1 Score** proposed in the original paper to measure the alignment with the reference API sequences.

**Results** The results are shown in Table 3. As observed, WorkflowLlama demonstrates strong OOD generalization performance on the T-Eval benchmark, despite being trained on different domains and tasks using different APIs. Notably, WorkflowLlama significantly outperforms the vanilla Llama3.1-8B as well as larger open-source models like Llama-2-70B and Qwen-72B, highlighting that fine-tuning with WorkflowBench enhances the model's out-of-distribution planning ability.

Table 3: Comparisons of F1 scores on the **PLAN** task of T-Eval. (**Bold** denotes the best score among models of the same category.)

| Model | F1 |
|---|---|
| **Proprietary Models** | |
| Claude2 | 84.9 |
| GPT-3.5 | 86.6 |
| GPT-4 | **86.7** |
| **Open-Source Models** | |
| Qwen-7B | 63.1 |
| Mistral-7B | 64.9 |
| Llama-3.1-8B | 68.2 |
| Qwen-14B | 69.7 |
| Llama-2-13B | 65.1 |
| Vicuna-13B | 54.0 |
| Baichuan2-13B | 52.1 |
| WizardLM-70B | 42.7 |
| Llama-2-70B | 63.1 |
| Qwen-72B | 73.4 |
| WorkflowLlama (8B) | **77.5** |

### 4.6 ABLATION STUDY

**Settings** To assess the efficacy of WorflowBench's components, we conduct an ablation study under the settings of unseen instructions (i.e., the ID setting).

**Results** Table 4 presents the performance results when the model is trained under different conditions: without synthetic data, without the task plan $\mathcal{P}$, without action-level comments $\mathcal{C}$, and without both $\mathcal{C}$ and $\mathcal{P}$. The experimental results reveal two key findings. **First**, the two types of natural language thoughts enhance the reasoning capabilities of the model. Removing either type of thought

Table 4: Ablation study results of Natural Language Thoughts on Workflow Orchestration (%).

| Model | CodeBLEU | | | | |
|---|---|---|---|---|---|
| | BLEU | Weighted N-Gram | AST | Data-Flow | Overall |
| WorkflowLlama | 9.4 | 11.1 | 55.1 | 38.0 | 39.3 |
| *w/o Task Plan* | 9.1 | 10.7 | 53.9 | 36.6 | 38.2 |
| *w/o Comment* | 9.1 | 10.8 | 54.9 | 35.3 | 38.1 |
| *w/o Task Plan & Comment* | 8.8 | 10.2 | 53.7 | 35.1 | 37.4 |
| *w/o Synthetic Data* | 7.8 | 9.4 | 53.5 | 35.4 | 37.3 |

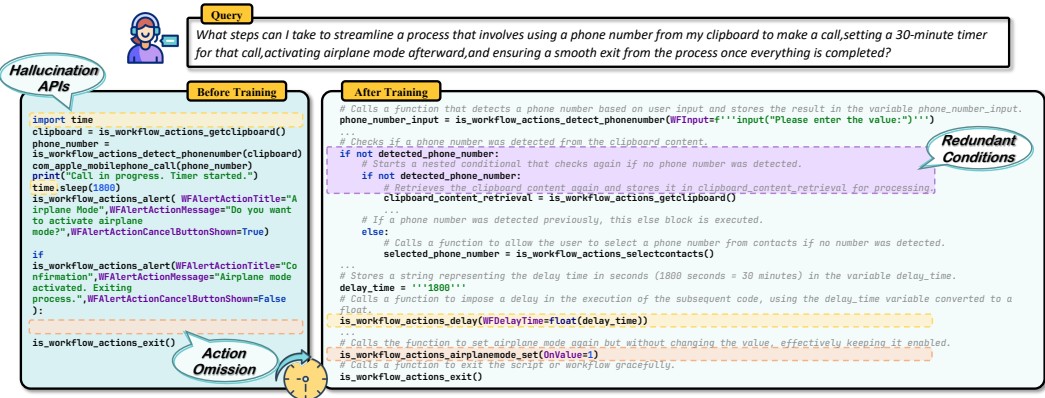

Figure 6: Case study of generated code between vanilla Llama-3.1-8B and WorkflowLlama.

leads to a decline in CodeBLEU performance. **Second**, training on large-scale synthetic data further improves performance, highlighting the effectiveness of the WorkflowBench expansion process.

## 4.7 CASE STUDY

To further illustrate the effect of fine-tuning on WorkflowBench, we present a typical example in Figure 6. In this case, the vanilla Llama-3.1 model exhibits two types of errors. **First**, the model does not adhere to the given instructions for workflow orchestration, using APIs outside the provided list, i.e., hallucination APIs. Specifically, it uses the time.sleep() function instead of is_workflow_actions_delay() to set a timer. **Second**, due to its relatively weak workflow orchestration capabilities, the model fails to complete all user instructions. Specifically, it does not activate airplane mode using the is_workflow_actions_airplanemode_set() function. Fine-tuning on WorkflowBench effectively alleviates these two issues. However, we observe that fine-tuning also introduces redundant actions. For instance, WorkflowLlama repeats the parsing check of the clipboard's content. We will address this redundancy problem in future work. To further demonstrate the large-scale node structure and complex logical framework of WorkflowBench, we also include a generated workflow code with long sequences in Appendix G.

## 5 CONCLUSION

In this paper, we present WorkflowBench to enhance the capability of large language models in workflow orchestration. By fine-tuning Llama-3.1-8B on WorkflowBench, we derive WorkflowLlama which can achieve superior performance on the workflow orchestration task exceeding all comparable baselines. Moreover, we adapt our WorkflowLlama on the T-Eval dataset and the experimental results reveal the generalization ability of our constructed WorkflowBench. However, WorkflowBench has some limitations, including the use of APIs limited to shortcuts and the absence of evaluation through actual execution (refer to Appendix H). Nevertheless, we believe that the dataset we have constructed holds significant potential to contribute to advancements in APA.

ACKNOWLEDGMENT

We sincerely thank all the anonymous reviewers for their valuable comments and constructive suggestions. This work was supported by The National Natural Science Foundation of China (No. 62376273), Beijing Nova Program (No. 20240484568), the Postdoctoral Fellowship Program of CPSF (Grant No. GZB20230343) and the China Postdoctoral Science Foundation (Grant No. 2023M741945).

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

## A  ALGORITHM OF TRANSCRIBING SHORTCUTS

---

**Algorithm 1:** Recursive Parsing of Property List to Construct Abstract Syntax Tree

---

**Data:** Shortcut file to be transcribed
**Result:** Abstract syntax tree of the actions
Initialize an empty tree with a root node and set `current_node` to root
**foreach** *action **in** action list* **do**
  Determine `action_type` and `mode` from action
  **if** `action_type` *is **Conditional*** **then**
    **HandleConditional**(`mode`, action)
  **else if** `action_type` *is **RepeatEach*** **then**
    **HandleLoop**(`mode`, action)
  **else if** `action_type` *is **RepeatCount*** **then**
    **HandleLoop**(`mode`, action)
  **else if** `action_type` *is **ChooseFromMenu*** **then**
    **HandleMatchCase**(`mode`, action)
  **else**
    **HandleDefault**(action)

**Function** *HandleConditional(`mode`, action)***:**
  **if** `mode` *== 0 (**start if**)* **then**
    **AddNode**(action)
    Set `current_node` to new node
  **else if** `mode` *== 1 (**else**)* **then**
    Move `current_node` to parent node
    **AddNode**(action)
    Set `current_node` to new node
  **else if** `mode` *== 2 (**end if**)* **then**
    Move `current_node` to parent node

**Function** *HandleLoop(`mode`, action)***:**
  **if** `mode` *== 0 (**start loop**)* **then**
    **AddNode**(action)
    Set `current_node` to new node
  **else if** `mode` *== 2 (**end loop**)* **then**
    Move `current_node` to parent node

**Function** *HandleMatchCase(`mode`, action)***:**
  **if** `mode` *== 0 (**start match**)* **then**
    **AddNode**(action)
    Set `current_node` to new node
  **else if** `mode` *== 1 (**start case**)* **then**
    **if** `current_node` *is match node* **then**
      **AddNode**(action)
      Set `current_node` to new node
    **else**
      Move `current_node` to parent match node
      **AddNode**(action)
      Set `current_node` to new node
  **else if** `mode` *== 2 (**end match**)* **then**
    Move `current_node` to parent node

**Function** *HandleDefault(action)***:**
  **AddNode**(action)

**Function** *AddNode(action)***:**
  Create new node with `action`
  Append new node to `current_node.children`
  Set parent of new node to `current_node`

---

## B PROMPT DESIGN

All of our prompts were manually designed and iteratively refined during early experiments.

### B.1 WORKFLOW ORCHESTRATION PROMPT

```
You are a very helpful AI assistant who can write corresponding
    Python main code based on user's query and usable Python
    function interface.

Please generate python main code based on the following query :
 {query}
You can start by using natural language to plan your tool call
    strategy, and then generate the complete code. For example, `
    Thought:
<tool call strategy>

Code:
```python
<main code>
````.
Note that your output should always include `Code:
```python
<main code>
````, formatted accordingly.
Here are some useful function interface you may use:
 {apis_docs}
```

### B.2 EVALUATOR PROMPT

```
You are a kindly code reviewer, I will provide you with a query, a
     list of allowed apis and a piece of code to be reviewed, you
    help me to check if the code to be reviewed is compliant with
    our specifications.
The requirements are as follows:
1. You **should return True even if the code implements additional
     functionality not required in the query**, as long as it
    roughly implements the requirements in the query.
2. We don't impose any requirements on code readability or naming
    conventions. You **should return True as long as the reviewed
    code doesn't use disallowed functions and reasonably
    accomplishes what is asked in the query in general terms**.
    There's no need to get strictly hung up on the details.
3. Return False if the code fails to fulfill the requirement in
    the query. e.g. if it is proposed in the query to turn down
    the battery level of the phone and the brightness of the
    screen, it is a failure to fulfill only any one of the
    functions.
4. Built-in python syntax such as `if`, `loop`, `input()`, and `
    print()` are allowed. Return False if the code uses **any
    external functions or apis** not in allowed apis list and not
    a built-in function such as input(), print(). For example, if
    I provide the is_workflow_openurl function, this should be
    used. Any use of any other library like requests etc. is a
    False.
query:{query}
list of allowed apis: {apis}
code to review: {code}
```

```
Your answer: [True or False with interpretation]
```

### B.3 COMMENT GENERATION PROMPT

```
A Shortcut is a sequence of actons, where each action is an API
    call, to execute user-provided queries.
As a user-friendly and patient assistant, your task is to provide
    a set of description of each line of the code scrippet. To
    save time, I have retrieved all the lines exclusive of blank
    lines of the code snippet and listed as a dictionary below the
     code.

Your answer should be in the json format as follows:
```json
{
    "line x": "<description-of-line-x>",
    "line x+1": "<description-of-line-x+1>",
    "...": "...",
    "line x+n": "<description-of-line-x+n>"
}```

The code is :
{code}
The lines are {lines}
```

### B.4 TASK PLAN GENERATION PROMPT

```
Based on this line by line description of the code, generate a
    flowchart of a workflow by natural language.
This is the code:
{code}
```

### B.5 TASK QUERY GENERATION PROMPT

```
As a helpful assistant, please help me craft a query. This query,
    formatted as a question, should describe the task a user wants
     to complete and adhere to the following criteria:
1. One of the solution to the task described in the query could be
     the python code below.
2. It should be close to real-world problems or requests.
3. It should include major parts of the code.
4. The query should not specify python.

For example, the code is:
{ICL_code}
And the expected output query should be similar to:
{ICL_query}

Please craft a query based on the examples and the following code:
{code}
```

### B.6 QUERY EXPANSION PROMPT

```
You are exceptionally skilled at crafting real-world user queries
    given some apis. Here are examples:{examples}. Please gain
    inspiration from the following api docs to create a high-
    quality realworld query.
Api docs for inspiration:
```python
{apis_string}
```
Please refer to the above examples and craft a new one!
Requirements: API name is strictly prohibited from appearing in
    the generated query. Each query should be complicated enough
    and can be solved using all apis above. The query **should be
    centered around {category} theme** and should not be spread
    out into unrelated pieces.
```

### B.7  QUALITY CONFIRMATION PROMPT

```
You are exceptionally skilled at polishing tool calling plan (i.e
    ., thought) and python code given a task.

Given task:
{query}

Old tool calling plan:
{thought}

 Old code:
{code}

 Used API doc:
{apis}

Here are examples for you to refer:{ICL_context}.
Please make sure the code is logically correct and operational.

Requirements:
[1] Ensure that both plan and code respond correctly to the task
    and that code calls match the plan, which you can do by
    tweaking, embellishing, and modifying both plan and code.
Plan does not have to be one-to-one correspondence of code; plan
    can be abbreviated.
[2] Please ensure that the code conforms to python syntax. Ensure
    that all python code is complete and runnable. You can add
    code when necessary.
[3] Every line of code should be preceded by a comment marked with
     a "#". When modifying the code, please modify the in-line
    comments accordingly.
[4] Ensure that all function parameter calls are correct and you
    can change the code in case of errors.
[5] Thought and code should be as concise while keeping the
    meaning intact.
[6] If there are cases including invalid binary code, replace them
     with reasonable text, delete them, or replace them with a
    reading operation on a file (especially when the binary code
    is an encoded image).
Respond strictly with JSON.
```

## B.8 VARIABLE RENAME PROMPT

```
You are a helpful assistant for renaming variable names in a code
    snippet.
The following code snippet is a part of a program, and variables
    are named in format 'variablex_'.
Your task is to rename these variables so that they conform to the
     programming specification and have some semantic meaning,
    which can be infered by relative function calls
And your output should only be a dictionary containing the old
    name-new name key value pair
The definition of some functions are not included, and you shouldn
    't modify them.
Following the code, there's a dictionary that contains short
    description of the uuid-named variable. And you can take it as
     reference.
Note that while the description might be the same, but the actual
    meaning is different across different variables. So you should
     not just copy the short description. Instead you'd better
    conprehensively consider the description, names of called
    functions, and the general logic.
The code is as follows:
{code}
The dictionary is as follows:
{description}
To save time, I have retrieved all the variables that requires to
    be renamed:
{variables}
```

## C QUALITY CONTROL PROCESS

The quality control process consists of three key steps aimed at ensuring the integrity and consistency of the generated code samples. First, GPT-4o-mini with in-context learning is employed to refine the workflows generated by the annotator. The primary objectives of this refinement are to minimize Python syntax errors, ensure that each line of code is accompanied by an explanatory comment, and eliminate meaningless binary encoding strings. Manual sampling was conducted to assess the effectiveness of this refinement, with results showing that 94.2% of the samples demonstrated improvements in at least one area, such as Python syntax, code clarity, and the correspondence between the code and the associated queries. The second step addresses the issue of empty code outputs, which arise when the input or output task plan is excessively lengthy. These instances are identified and excluded from the dataset, leading to the removal of approximately 3% of the total samples. The final step involves ensuring consistency with expected function and API definitions by using the Python interpreter to automatically detect syntax errors. Test functions are constructed according to the API specifications, and any samples that fail to execute correctly due to violations of parameter constraints are discarded. This step results in the exclusion of approximately 18% of the samples from the dataset.

## D ANALYSIS OF API RETRIEVER

In our main experiments, the primary objective is to demonstrate that WorkflowLlama can generalize effectively when faced with Out-Of-Distribution (OOD) APIs, i.e., APIs that were not seen during training. To further validate the robustness of WorkflowLlama, we conduct experiments in which golden APIs are not provided during inference. In the workflow orchestration task, a single query may correspond to dozens of APIs. Selecting the correct APIs for a given task requires advanced reasoning capabilities, and simple models such as dual-tower semantic matchers (Gao et al., 2021) may fail to capture the nuances necessary for accurate API selection. Therefore, we use in-context

learning (ICL) to prompt large language models (LLMs) to extract the appropriate APIs for a query. The specific prompt used for API retrieval is as follows:

```
You are an API retriever capable of identifying the APIs required
    to complete a query, based on both the user query and an API
    list.

API list:
\{APIs with descriptions\}

Here are some examples:
\{example queries and API mappings\}

For the following query, which APIs are needed to complete the
    task? Please return a list of required APIs without any
    explanations.

query: \{query\}
APIs:
```

To enhance the capabilities of API Retriever, we select the most 50 similar queries from the training set using the MiniLM-L6-v2 model (Wang et al., 2020) to generate appropriate ICL samples. The performance of the API retriever is evaluated based on precision and recall, and the results are listed in Tabel 5. Although the retriever's precision and recall are not particularly high, manual inspection of 50 randomly selected cases revealed that 84% of the retrieved APIs successfully fulfilled the tasks specified in the queries.

| API Retriever | Precision | Recall |
|---|---|---|
| GPT-4o-mini | **42.5%** | 36.4% |
| Qwen2.5-72B | 40.6% | **40.7%** |

Table 5: Precision and Recall for different API retrievers.

| API Retriever | Pass Rate |
|---|---|
| Golden | **70.4%** |
| GPT-4o-mini | 66.7% |
| Qwen2.5-72B | 69.0% |

Table 6: Pass Rate of WorkflowLlama with different API retrievers.

To further validate this insight, we conduct workflow orchestration experiments using APIs retrieved by LLMs in the OOD setting. The results of these experiments, comparing the pass rates of WorkflowLlama using different API sources, are summarized in Table 6. The results show that WorkflowLlama, trained with golden APIs, experiences only a minor performance decrease when using APIs retrieved by LLMs. Notably, even when APIs are retrieved by the open-source model Qwen2.5 (Team, 2024), WorkflowLlama achieves comparable or better performance than GPT-4o-mini.

In conclusion, the main takeaway from these additional experiments is that WorkflowLlama remains effective in OOD settings, even in the absence of golden APIs during inference. This emphasizes the flexibility and robustness of our framework for automating workflow orchestration in real-world scenarios, where APIs may not always be available at training time.

## E  CASE STUDY OF SHORTCUTS

We provide a real-world shortcut example, which includes the following three presentation forms: the raw property list configuration file, the Python code after transcription and variable renaming, and the visual interface on MacOS.

The raw property list configuration file is presented below. For the sake of brevity, we have omitted the middle portion containing the actions.

```
{
  "WFWorkflowClientVersion": "1050.19",
  "WFWorkflowInputContentItemClasses": [
```

```
      "WFContactContentItem",
      "WFPhoneNumberContentItem",
      "WFRichTextContentItem",
      "WFStringContentItem"
    ],
    "WFWorkflowClientRelease": "3.0",
    "WFWorkflowMinimumClientVersion": 900,
    "WFWorkflowIcon": {
      "WFWorkflowIconStartColor": 4292093695,
      "WFWorkflowIconGlyphNumber": 59814
    },
    "WFWorkflowImportQuestions": [
      {
        "ParameterKey": "WFCallContact",
        "Category": "Parameter",
        "ActionIndex": 12,
        "Text": "PreSetup CallNumber for FastStart.\n\n"
      },
      {
        "ParameterKey": "WFTextActionText",
        "Category": "Parameter",
        "ActionIndex": 13,
        "Text": "The call will be Ended after 30min! (1800 sec
        ↪   )\n\n\n",
        "DefaultValue": "1800"
      }
    ],
    "WFWorkflowActions": [
      {
        "WFWorkflowActionIdentifier":
        ↪   "is.workflow.actions.detect.phonenumber",
        "WFWorkflowActionParameters": {
          "WFInput": {
            "Value": {
              "string": " ",
              "attachmentsByRange": {
                "{0, 1}": {
                  "Type": "ExtensionInput"
                }
              }
            },
            "WFSerializationType": "WFTextTokenString"
          },
          "UUID": "ECAE88CB-B8C3-4E5C-9781-4BC35F89AB0C"
        }
      },
      {
        "WFWorkflowActionIdentifier":
        ↪   "is.workflow.actions.getclipboard",
        "WFWorkflowActionParameters": {
          "UUID": "7379BE67-847D-4F9F-8810-A6371C6CEF79"
        }
      },
      {
        "WFWorkflowActionIdentifier":
        ↪   "is.workflow.actions.detect.phonenumber",
        "WFWorkflowActionParameters": {
          "WFInput": {
            "Value": {
```

```
          "string": " ",
          "attachmentsByRange": {
            "{0, 1}": {
              "OutputUUID":
              ↪    "7379BE67-847D-4F9F-8810-A6371C6CEF79",
              "Type": "ActionOutput",
              "OutputName": "Zwischenablage"
            }
          }
        },
        "WFSerializationType": "WFTextTokenString"
      },
      "UUID": "F3E902F4-1A98-43C8-9D60-C4E52B372E27"
    }
  },
  {
    "WFWorkflowActionIdentifier":
    ↪    "is.workflow.actions.conditional",
    "WFWorkflowActionParameters": {
      "WFInput": {
        "Type": "Variable",
        "Variable": {
          "Value": {
            "OutputUUID":
            ↪    "F3E902F4-1A98-43C8-9D60-C4E52B372E27",
            "Type": "ActionOutput",
            "OutputName": "Telefonnummern"
          },
          "WFSerializationType": "WFTextTokenAttachment"
        }
      },
      "WFControlFlowMode": 0,
      "GroupingIdentifier":
      ↪    "CF250547-0705-4916-A3E6-C7DB85FD8C7B",
      "WFCondition": 101
    }
  }, ...
  {
    "WFWorkflowActionIdentifier": "is.workflow.actions.exit",
    "WFWorkflowActionParameters": {}
  }
],
"WFWorkflowMinimumClientVersionString": "900",
"WFWorkflowTypes": [
  "WatchKit",
  "ActionExtension",
  "NCWidget"
]
}
```

It can be observed that this configuration file employs non-semantic hexadecimal strings to represent variables and uses keywords such as `is.workflow.actions.conditional` and `GroupingIdentifier` to implement logic controls like conditions, making it inherently difficult to read and comprehend. Consequently, we have converted it into a Python-like code format. The Python code, after transcription, variable renaming, and commenting, is shown as follows:

```python
# Calls a function that detects a phone number based on user input and
↪    stores the result in the variable phone_number_input.
phone_number_input = is_workflow_actions_detect_phonenumber(
↪    WFInput=f'''input("Please enter the value: ")''')
```

```python
# Calls a function to retrieve the current content from the clipboard and
↪  stores it in the variable clipboard_content.
clipboard_content = is_workflow_actions_getclipboard()
# Detects a phone number from the clipboard content and assigns the
↪  result to detected_phone_number.
detected_phone_number = is_workflow_actions_detect_phonenumber(
↪  WFInput=f'''{clipboard_content}''')
# Checks if a phone number was detected from the clipboard content.
if not detected_phone_number:
    # Starts a nested conditional that checks again if no phone number
    ↪  was detected.
    if not detected_phone_number:
        # Retrieves the clipboard content again and stores it in
        ↪  clipboard_content_retrieval for processing.
        clipboard_content_retrieval = is_workflow_actions_getclipboard()
        # Calls a function to convert the retrieved clipboard content
        ↪  into a phone number and assigns it to selected_phone_number.
        selected_phone_number = is_workflow_actions_phonenumber(
        ↪  WFPhoneNumber=clipboard_content_retrieval)
    # If a phone number was detected previously, this else block is
    ↪  executed.
    else:
        # Calls a function to allow the user to select a phone number
        ↪  from contacts if no number was detected.
        selected_phone_number = is_workflow_actions_selectcontacts()
# Displays an alert message to the user indicating that the call will end
↪  after 30 minutes.
is_workflow_actions_alert( WFAlertActionMessage='''
# Sets the title of the alert to inform the user they are starting a call
↪  to the selected phone number.
The Call ends after 30 min''', WFAlertActionTitle=f'''Start calling
↪  "{coerce_variable(value=selected_phone_number,
↪  coercion_class="WFPhoneNumberContentItem")}"''')
# Initiates a call to the selected phone number using a specific function
↪  to handle mobile phone calls.
com_apple_mobilephone_call( WFCallContact=selected_phone_number)
# Stores a string representing the delay time in seconds (1800 seconds =
↪  30 minutes) in the variable delay_time.
delay_time = '''1800'''
# Calls a function to impose a delay in the execution of the subsequent
↪  code, using the delay_time variable converted to a float.
is_workflow_actions_delay( WFDelayTime=float(delay_time))
# Activates airplane mode by calling a function with OnValue set to 1 to
↪  enable it, and operation set indicating that it should be set.
is_workflow_actions_airplanemode_set( OnValue=1, operation='''set''')
# Calls a delay function again with a specified delay time of 3 seconds
↪  before proceeding.
is_workflow_actions_delay( WFDelayTime=3.0)
# Calls the function to set airplane mode again but without changing the
↪  value, effectively keeping it enabled.
is_workflow_actions_airplanemode_set( OnValue=1)
# Calls a function to exit the script or workflow gracefully.
is_workflow_actions_exit()
```

We can clearly see that the transcribed Python code is of high quality, with strong readability, making it well-suited for training LLMs that have been pre-trained on extensive Python code.

For easy understanding, we also present the visual interface of this shortcut in Figure 7, which corresponds to the transcribed Python code on a line-by-line basis.

## F    CASE STUDY OF WORKFLOWBENCH

In this section, we provide a workflow example of WorkflowBench based on the shortcut listed in Appendix E. We omit the annotated workflow code that has been shown in Appendix E.

The query $\mathcal{Q}$ is:

```
What steps can I take to streamline a process that involves using
    a phone number from my clipboard to make a call, setting a 30-
    minute timer for that call, activating airplane mode afterward
    , and ensuring a smooth exit from the process once everything
    is completed?
```

The list of provided APIs is:

```
[
    "is.workflow.actions.selectcontacts",
    "com.apple.mobilephone.call",
    "is.workflow.actions.exit",
    "is.workflow.actions.delay",
    "is.workflow.actions.alert",
    "is.workflow.actions.phonenumber",
    "is.workflow.actions.detect.phonenumber",
    "is.workflow.actions.airplanemode.set",
    "is.workflow.actions.getclipboard"
]
```

The task plan $\mathcal{P}$ is:

```
1. **Start**
   - Begin the process.
2. **Get User Input**
   - Prompt the user: "Please enter the value:"
   - Call function `is_workflow_actions_detect_phonenumber` to
     detect a phone number from input.
   - Store the result in `phone_number_input`.
3. **Get Clipboard Content**
   - Call function `is_workflow_actions_getclipboard` to retrieve
     current clipboard content.
   - Store it in `clipboard_content`.
4. **Detect Phone Number from Clipboard**
   - Call `is_workflow_actions_detect_phonenumber` with `
     clipboard_content` to detect a phone number.
   - Store the result in `detected_phone_number`.
5. **Check if Phone Number Detected**
   - If `detected_phone_number` is not found:
     1. **Retrieve Clipboard Again**
        - Call `is_workflow_actions_getclipboard` and store result
          in `clipboard_content_retrieval`.
     2. **Convert Clipboard to Phone Number**
        - Call `is_workflow_actions_phonenumber` with `
          clipboard_content_retrieval`.
        - Store the result in `selected_phone_number`.
   - Else:
     1. **Select from Contacts**
        - Call `is_workflow_actions_selectcontacts` to select a
          phone number.
        - Store it in `selected_phone_number`.
6. **Alert User**
   - Call `is_workflow_actions_alert` to display an alert
     indicating the call will end after 30 minutes.
```

```
    1. **Set Alert Message**
       - Alert message: "The Call ends after 30 min".
       - Alert title: "Start calling [selected_phone_number]".
7. **Initiate Call**
   - Call `com_apple_mobilephone_call` with `selected_phone_number
     ` to start the call.
8. **Set Delay Time**
   - Store "1800" in `delay_time`.
   - Call `is_workflow_actions_delay` with `float(delay_time)` to
     impose a 30-minute delay.
9. **Enable Airplane Mode**
   - Call `is_workflow_actions_airplanemode_set` with `OnValue=1`
     (to enable).
   - Call `is_workflow_actions_delay` with a delay of 3 seconds.
10. **Keep Airplane Mode Enabled**
    - Call `is_workflow_actions_airplanemode_set` with `OnValue=1`
      to maintain airplane mode.
11. **Exit Workflow**
    - Call `is_workflow_actions_exit` to terminate the script
      gracefully.
12. **End**
    - The process completes.
```

## G  CASE STUDY OF GENERATED WORKFLOWS BY WORKFLOWLLAMA

Task query:

```
What steps would I need to take to develop an automated messaging
    system that alerts users when their message bank is depleted
    and enables them to dispatch pre-defined messages?
    Additionally, how can this system monitor for updates to its
    automation script and notify users accordingly?
```

The list of provided APIs is:

```
[
    "is_workflow_actions_notification",
    "is_workflow_actions_setvariable",
    "is_workflow_actions_delay",
    "is_workflow_actions_sendmessage",
    "is_workflow_actions_openurl",
    "is_workflow_actions_downloadurl",
    "is_workflow_actions_contacts",
    "is_workflow_actions_alert",
    "is_workflow_actions_text_split",
    "is_workflow_actions_count",
    "is_workflow_actions_nothing",
    "is_workflow_actions_comment",
    "is_workflow_actions_getwebpagecontents",
    "is_workflow_actions_exit",
    "is_workflow_actions_documentpicker_open",
    "is_workflow_actions_getipaddress",
    "is_workflow_actions_documentpicker_save",
    "is_workflow_actions_getitemfromlist",
    "is_workflow_actions_math",
    "is_workflow_actions_url"
]
```

The generated task plan $\mathcal{P}$ is:

```
1. **Start**
   - Initialize the process.
2. **Define Metadata**
   - Create a dictionary named `message_bank_metadata` containing:
     - Name: "Message Bank"
     - Author: "@ar0512"
     - Version: "1.0"
     - Last Updated: "01/17/2020"
     - Change Log: "Initial release"
     - RoutineHub ID: "3967"
3. **Assign Metadata to Variable**
   - Set `Info` equal to `message_bank_metadata`.
4. **Retrieve Contacts**
   - Call `is_workflow_actions_contacts()` to get a list of
      contacts and store it in `contacts_list`.
...
11. **End**
    - Complete the workflow.
```

The generated workflow code $\mathcal{A}$ is:

```
# Defines a dictionary named message_bank_metadata that contains
    metadata about the message bank including its name, author,
    version, last updated date, change log, and RoutineHub ID.
message_bank_metadata = {'''Name''': '''Message Bank''', '''Author
    ''': '''@ar0512''', '''Version''': '''1.0''', '''Last Updated
    ''': '''01/17/2020''', '''Change Log''': {01/17/2020: Initial
    release}, '''RoutineHub ID''': '''3967'''}
# Assigns the message_bank_metadata dictionary to a variable named
     Info for easier reference later in the code.
Info = message_bank_metadata
# Calls a function is_workflow_actions_contacts() that retrieves a
     list of contacts and assigns it to contacts_list.
contacts_list = is_workflow_actions_contacts()
# Prompts the user for input and stores the value entered in
    user_input_value.
user_input_value = f'''input("Please enter the value: ")'''
# Counts the number of items in user_input_value using the
    is_workflow_actions_count function and stores the result in
    item_count.
item_count = is_workflow_actions_count( WFCountType='''Items''',
    Input=user_input_value)
# Checks if item_count is greater than 0 to proceed with further
    actions.
if item_count > 0.0:
    # If the condition is true, prompts the user for input again
        and stores the value in user_input_value_2.
    user_input_value_2 = f'''input("Please enter the value: ")'''
    # Counts the number of items in user_input_value_2 and stores
        the result in item_count_2.
    item_count_2 = is_workflow_actions_count( WFCountType='''Items
        ''', Input=user_input_value_2)
    # Checks if item_count_2 is greater than 1 to determine if the
         user has entered multiple values.
    if item_count_2 > 1.0:
        # If the condition is true, prompts the user for input
            again and stores the value in user_input_value_3.
        user_input_value_3 = f'''input("Please enter the value: ")
            '''
```

```
# Counts the number of items in user_input_value_3 and
    stores the result in item_count_3.
item_count_3 = is_workflow_actions_count( WFCountType='''
    Items''', Input=user_input_value_3)
# Checks if item_count_3 is greater than 2 to confirm the
    user has entered enough values for processing.
if item_count_3 > 2.0:
    # Displays an alert to the user indicating the maximum
        number of messages that can be sent at a time.
    is_workflow_actions_alert( WFAlertActionMessage='''You
        may send a maximum of 10 messages at a time.
        Please choose your message.''', WFAlertActionTitle
        ='''Maximum Number of Messages''',
        WFAlertActionCancelButtonShown=False)
    # Prompts the user to choose a message from the
        contacts list and stores the chosen message in
        user_input_message.
    user_input_message = f'''input("Please choose your
        message: ")'''
    # Splits the text of the chosen message into parts and
        stores the result in split_text.
    split_text = is_workflow_actions_text_split( text=
        user_input_message)
    # Retrieves the message to send based on the user's
        choice from split_text and stores it in
        message_to_send.
    message_to_send = is_workflow_actions_getitemfromlist(
        WFInput=split_text, WFItemIndex='''2''',
        WFItemSpecifier='''Item At Index''')

... (134 lines of code)
```

## H  LIMITATIONS

While the framework proposed in this paper represents notable progress in workflow orchestration, it also has certain limitations that warrant discussion. First, the APIs used in our work are exclusively derived from Apple Shortcuts application, resulting in a lack of coverage across more diverse fields and potentially limiting the generalizability of the dataset to broader application contexts. Second, our approach lacks evaluation through actual execution. This limitation arises due to the complexities involved in executing workflows, such as the need for intricate user registration and permission acquisition. Moreover, the APIs are subject to frequent changes as applications continue to evolve, making it challenging to implement a consistent executable evaluation. Consequently, our evaluation is limited to static analysis.

## I  ETHICAL STATEMENT

In this study, the dataset construction process was fully automated using LLMs and algorithms for data annotation, eliminating the need for human annotators and thereby avoiding concerns related to annotator compensation and working conditions. The data utilized was collected through web scraping from publicly accessible sources, with strict adherence to the Terms of Service (ToS) of the respective websites. Scraping was avoided on platforms where such activity is explicitly prohibited, ensuring compliance with ethical standards. Additionally, no personally identifiable information (PII) or private user data was collected at any stage of the research process. All data was anonymized to protect privacy and mitigate any potential ethical concerns related to user information.

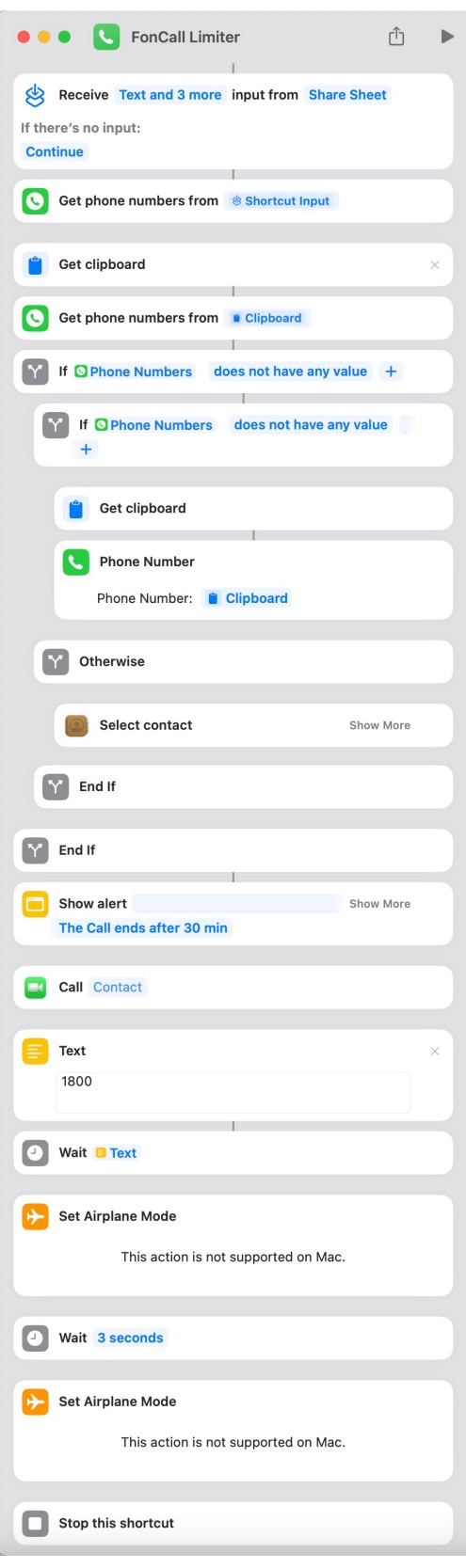

Figure 7: The visual interface of the shortcut RoutineHub • CallDuration Limite.

