# OpenReview forum: "WorkflowLLM: Enhancing Workflow Orchestration Capability of Large Language Models"
_ICLR.cc/2025/Conference — ICLR 2025 Poster_

### Official Review · Reviewer_mrzv · 2024-10-31

**Soundness:** 2
**Presentation:** 2
**Contribution:** 1
**Rating:** 6
**Confidence:** 4

**Summary:**

This paper proposes a pipeline for constructing workflow (RPA) data to enhance LLMs' workflow orchestration capabilities. The core motivation stems from the observation that real-world workflows require more complex logic and longer sequences than what current LLMs can directly generate. The primary contribution is the construction of a dataset containing over 100K samples. The main limitations lie in the lack of certain technical details (such as specific ChatGPT versions used for annotation) and the lack of deeper investigation into model capability improvements.

**Strengths:**

1. Accurate and valuable problem identification. The paper identifies the limitations of existing work in handling real-world workflows: difficulty in processing long-sequence workflows with complex logical relationships like control flow. This observation directly addresses practical application pain points, making the motivation very solid.
2. Proposes a systematic data construction method. The paper designs a complete data synthesis process: from real data collection, workflow abstract representation, data synthesis to quality control, with detailed design considerations at each stage. The resulting large-scale dataset provides an important resource for this field.
3. Comprehensive experimental validation. The effectiveness of the method is strongly supported through multi-dimensional evaluation using CodeBLEU and Pass Rate, cross-domain testing with T-Eval, along with detailed case analysis and ablation studies.

**Weaknesses:**

1. Important technical details are missing. The paper mentions using ChatGPT for data annotation and optimization multiple times but doesn't specify the model version. For work with data construction as its core contribution, this seriously affects reproducibility and rigor.
2. Lacks methodological innovation and mechanism analysis. Although data construction is an important contribution, the paper lacks in-depth analysis of how this data enhances model capabilities. Specifically, it doesn't investigate whether the improvements come from enhanced logical reasoning abilities or simply better format matching. Without such analysis, it's difficult to determine if the model has truly learned to reason about complex workflows or has merely memorized patterns from the training data.
3. Missing critical ablation experiments. The paper doesn't compare the performance difference between the Annotator Model and WorkflowLLM, despite both models using the same training method and data type. This results in question the necessity of the data synthesis strategy and weakens the core argument.
4. The paper mentions that the Annotator Model can generate workflows with over 70 actions, and theoretically WorkflowLLM should be capable of the same. Given the general limitations of LLMs in long sequence generation, including such long-sequence workflows in the appendix would further demonstrate the work's contribution.
5. Given that the core contribution is data construction, whether the dataset will be open-sourced directly affects the practical value of the work. The authors should consider open-sourcing the dataset to promote development in this field.

**Questions:**

1. The performance of generic LLMs in workflow orchestration through agent-like approaches would serve as an interesting baseline for comparison. Through this comparison, we could better understand whether the core challenge lies in the model's reasoning capabilities or in its ability to maintain consistent output formats. This point won't affect the score; it's just for discussion.

---

> ### Author Response · Authors · 2024-11-22
> **We used gpt-4o-mini-2024-07-18 and gpt-4o-2024-08-06 throughout this paper.**
>
> **Q1:** Important technical details are missing. The paper mentions using ChatGPT for data annotation and optimization multiple times but doesn't specify the model version. For work with data construction as its core contribution, this seriously affects reproducibility and rigor.
>
> **A1:** Apologies for the misunderstanding caused to readers. We will revise the paper to clarify the parts involving ChatGPT by replacing them with GPT-4o-mini and GPT-4o, with annotations provided in footnotes. The GPT versions we used are `gpt-4o-mini-2024-07-18` and `gpt-4o-2024-08-06`.

---

> ### Author Response · Authors · 2024-11-22
> **WorkflowLlama outperforms GPT-4o with ICL under an evaluation method that is entirely format-agnostic.**
>
> **Q2**: The paper lacks in-depth analysis of how this data enhances model capabilities. Specifically, it doesn't investigate whether the improvements come from enhanced logical reasoning abilities or simply better format matching. Without such analysis, it's difficult to determine if the model has truly learned to reason about complex workflows or has merely memorized patterns from the training data.
>
> **A2**: To mitigate the influence of formatting, we adopted a **Pass Rate** evaluation method based on GPT-4o-mini. The specific evaluation prompt is provided in `Appendix B.2`. We want to emphasize that **this evaluation is entirely format-independent, focusing solely on whether the task was completed using the given API, and achieving 81.2% agreement with human evaluations (Section 4.2)**.  As demonstrated in the examples below, even though the candidate code having significant differences from the Reference Code in format (only acquring 11% CodeBLEU score), successfully completes the task and is therefore judged as `True` by GPT-4o-mini.
> Under this evaluation method, which is entirely agnostic to formatting, WorkflowLlama surpasses GPT-4o with ICL by **9.3%** and **9.0%** in pass rate for ID and OOD settings, respectively.
>
> To further validate our model's advantages, we manually annotated **all 1190 test instances** of WorkflowLlama and GPT-4o with In-Context Learning (ICL). The experimental results are as follows:
> | Model                     | Human Pass Rate       |
> |---------------------------|-----------------------|
> | GPT-4o with ICL           | 65.9%                 |
> | WorkflowLlama              | **71.1%**             |
>
> These results demonstrate that WorkflowLlama has truly learned to reason about complex workflows, rather than merely memorizing patterns from the training data.
>
> ---
> ### Case Study
>
> **Query:**
> What steps would I need to follow to develop a script that allows a user to select specific YouTube videos corresponding to various amusement park rides?
>
> **Reference Code:**
> ```python
> # Initiates pattern matching on the function that fetches user input.
> match input():
>     # Defines a case for when the input is 'Blast Off'.
>     case "Blast Off":
>         # Sets the 'blast_off_url' variable to a specific YouTube video URL when input matches 'Blast Off'.
>         blast_off_url = is_workflow_actions_openurl( Show-WFInput=True, WFInput='''https://www.youtube.com/watch?v=nKOtGJECa_c''')
>     # Defines a case for when the input is 'Boomerang'.
>     case "Boomerang":
>         # Sets the 'boomerang_url' variable to a specific URL when input matches 'Boomerang'.
>         boomerang_url = is_workflow_actions_openurl( Show-WFInput=True, WFInput='''https://www.youtube.com/watch?v=bTK0ymzGV4g''')
>     # Defines a case for when the input is 'Mind Eraser'.
>     case "Mind Eraser":
>         # Sets the 'mind_eraser_url' variable to a URL when input matches 'Mind Eraser'.
>         mind_eraser_url = is_workflow_actions_openurl( Show-WFInput=True, WFInput='''https://www.youtube.com/watch?v=VXW_8N1Q-TM''')
> ...(more code)
> ```
>
> **Candidate Code (CodeBLEU: 11%, BLEU: 0.2%, weighted N-gram: 0.2%, AST: 16.1%, dataflow:13.1%, GPTEval: True):**
> ```python
> # Prompt the user to enter the amusement park ride name.
> Input = f'{input("Please enter the amusement park ride: ")}'
>
> # Construct the YouTube search URL by appending the user input to the base URL.
> search_url = f'https://www.youtube.com/results?search_query={Input}'
>
> # Open the constructed YouTube search URL in Safari.
> is_workflow_actions_openurl(WFInput=search_url)
> ```
> ---

---

> ### Author Response · Authors · 2024-11-22
> **WorkflowLLM outperforms the Annotator Model.**
>
> **Q3:** Missing critical ablation experiments. The paper doesn't compare the performance difference between the Annotator Model and WorkflowLLM, despite both models using the same training method and data type. This results in questioning the necessity of the data synthesis strategy and weakens the core argument.
>
> **A3:**
> Thank you for your concern about the ablation experiment. In fact, we have included this comparison in `Table 4` of the paper. The "without Synthetic Data" setting essentially corresponds to the Annotator Model. As shown in Table 4, **WorkflowLLM with synthetic data outperforms the Annotator Model**, demonstrating the necessity of the data synthesis strategy.

---

> ### Author Response · Authors · 2024-11-22
> **Case study of long-sequence workflows.**
>
> **Q4:** The paper mentions that the Annotator Model can generate workflows with over 70 actions, and theoretically WorkflowLLM should be capable of the same. Given the general limitations of LLMs in long sequence generation, including such long-sequence workflows in the appendix would further demonstrate the work's contribution.
>
> **A4:**  Thank you for your valuable comments. We will put the following examples in the appendix.
>
> ---
> ### Case study of long-sequence workflows
>
> **Task Query**:
>  What steps would I need to take to develop an automated messaging system that alerts users when their message bank is depleted and enables them to dispatch pre-defined messages? Additionally, how can this system monitor for updates to its automation script and notify users accordingly?
>
>
> **Workflow Code**:
> ```python
> # Defines a dictionary named message_bank_metadata that contains metadata about the message bank including its name, author, version, last updated date, change log, and RoutineHub ID.
> message_bank_metadata = {'''Name''': '''Message Bank''', '''Author''': '''@ar0512''', '''Version''': '''1.0''', '''Last Updated''': '''01/17/2020''', '''Change Log''': {01/17/2020: Initial release}, '''RoutineHub ID''': '''3967'''}
> # Assigns the message_bank_metadata dictionary to a variable named Info for easier reference later in the code.
> Info = message_bank_metadata
> # Calls a function is_workflow_actions_contacts() that retrieves a list of contacts and assigns it to contacts_list.
> contacts_list = is_workflow_actions_contacts()
> # Prompts the user for input and stores the value entered in user_input_value.
> user_input_value = f'''input("Please enter the value: ")'''
> # Counts the number of items in user_input_value using the is_workflow_actions_count function and stores the result in item_count.
> item_count = is_workflow_actions_count( WFCountType='''Items''', Input=user_input_value)
> # Checks if item_count is greater than 0 to proceed with further actions.
> if item_count > 0.0:
>     # If the condition is true, prompts the user for input again and stores the value in user_input_value_2.
>     user_input_value_2 = f'''input("Please enter the value: ")'''
>     # Counts the number of items in user_input_value_2 and stores the result in item_count_2.
>     item_count_2 = is_workflow_actions_count( WFCountType='''Items''', Input=user_input_value_2)
>     # Checks if item_count_2 is greater than 1 to determine if the user has entered multiple values.
>     if item_count_2 > 1.0:
>         # If the condition is true, prompts the user for input again and stores the value in user_input_value_3.
>         user_input_value_3 = f'''input("Please enter the value: ")'''
>         # Counts the number of items in user_input_value_3 and stores the result in item_count_3.
>         item_count_3 = is_workflow_actions_count( WFCountType='''Items''', Input=user_input_value_3)
>         # Checks if item_count_3 is greater than 2 to confirm the user has entered enough values for processing.
>         if item_count_3 > 2.0:
>             # Displays an alert to the user indicating the maximum number of messages that can be sent at a time.
>             is_workflow_actions_alert( WFAlertActionMessage='''You may send a maximum of 10 messages at a time. Please choose your message.''', WFAlertActionTitle='''Maximum Number of Messages''', WFAlertActionCancelButtonShown=False)
>             # Prompts the user to choose a message from the contacts list and stores the chosen message in user_input_message.
>             user_input_message = f'''input("Please choose your message: ")'''
>             # Splits the text of the chosen message into parts and stores the result in split_text.
>             split_text = is_workflow_actions_text_split( text=user_input_message)
>             # Retrieves the message to send based on the user's choice from split_text and stores it in message_to_send.
>             message_to_send = is_workflow_actions_getitemfromlist( WFInput=split_text, WFItemIndex='''2''', WFItemSpecifier='''Item At Index''')
>
> ... (134 lines of code)
>
> ```

---

> ### Author Response · Authors · 2024-11-22
> **We have re-uploaded our code and detailed document.**
>
> **Q5:** Given that the core contribution is data construction, whether the dataset will be open-sourced directly affects the practical value of the work. The authors should consider open-sourcing the dataset to promote development in this field.
>
> **A5:**
> Thank you for pointing it out.
> We have uploaded our code and detailed document to an **anonymized** repository: `https://anonymous.4open.science/r/WorkflowLLM-3E5D`. Please refer to `READEME.md` for details.
>
> ---
>
> Update:
>
> We update the anonymous url.

---

> ### Author Response · Authors · 2024-11-22
> **Agent-like approaches are promising future directions for workflow construction.**
>
> **Q6**: The performance of generic LLMs in workflow orchestration through agent-like approaches would serve as an interesting baseline for comparison. Through this comparison, we could better understand whether the core challenge lies in the model's reasoning capabilities or in its ability to maintain consistent output formats. This point won't affect the score; it's just for discussion.
>
> **A6:**
>
> Thank you for your insightful question regarding workflow construction paradigms. Below, we provide our thoughts on this topic while addressing your concerns:
>
> 1. The primary objective of this paper is to enhance the ability of LLMs in static scenarios for workflow orchestration. Therefore, approaches like ReAct [1] have not been considered as baselines in this study.
>
> 2. The workflows in our study contain an average of **more than 70 actions**, much more than ever before. Methods like ReAct [1] and Reflexion [2] focus on dynamic environments and perform well on tasks with 10–20 steps [3, 4]. However, these methods currently lack consistency when handling tasks with a higher number of steps and more complex tasks [5].
>
> Therefore, we believe that adapting agent-based methods like ReAct to workflow generation holds significant potential but presents non-trivial challenges. Thus, we propose this as a direction for future work.
>
> ---
> **References**
>
> [1] Yao, Shunyu, et al. "ReAct: Synergizing Reasoning and Acting in Language Models." *The Eleventh International Conference on Learning Representations*.
>
> [2] Shinn, Noah, et al. "Reflexion: Language agents with verbal reinforcement learning." *Advances in Neural Information Processing Systems 36* (2024).
>
> [3] Shridhar, Mohit, et al. "ALFWorld: Aligning Text and Embodied Environments for Interactive Learning." *International Conference on Learning Representations*.
>
> [4] Yao, Shunyu, et al. "Webshop: Towards scalable real-world web interaction with grounded language agents." *Advances in Neural Information Processing Systems 35* (2022): 20744-20757.
>
> [5] Xie, Jian, et al. "TravelPlanner: A Benchmark for Real-World Planning with Language Agents." *Forty-first International Conference on Machine Learning*.

---

> ### Author Response · Authors · 2024-11-22
> **Follow-Up: Seeking Further Feedback**
>
> Dear Reviewer,
>
> I hope you're doing well. Following up on our recent exchange regarding this paper, I wanted to check if there are any further concerns or feedback from your side. Your insights are invaluable to us, and we're keen to address any remaining issues.

---

> ### Author Response · Authors · 2024-11-25
> **Looking for Further Discussion and Feedback**
>
> **Dear Reviewer mrzv**,
>
> In the above responses, we have try our best to answer your questions and solve your confusions. Due to the rebuttal ddl is coming, we are very willing to have a more in-depth discussion with you, and we welcome you to give us more suggestions. If you have additional suggestions, please let us know and we will try to respond as quickly as possible.

---

> ### Author Response · Authors · 2024-11-26
> **Follow-up on Rebuttal Discussion and Further Feedback**
>
> Dear Reviewer mrzv,
>
> I hope this message finds you well. We appreciate the time and effort you have put into reviewing our paper and providing valuable feedback. As the rebuttal deadline is fast approaching, we would like to kindly follow up and see if there are any remaining points or concerns that we could address.
>
> We are eager to engage in further discussion and would greatly appreciate any additional thoughts or suggestions you might have. Please do not hesitate to share any further feedback, and we will ensure a prompt response.
>
> Thank you once again for your time and consideration.
>
> Best regards,
>
> Authors of Submission 2616

---

> ### Author Response · Authors · 2024-11-27
> **Kind Request for Your Feedback Before the Rebuttal Deadline**
>
> Dear Reviewer mrzv,
>
> We have diligently addressed all the concerns raised during the rebuttal period, and we would like to kindly ask if any remaining issues have been resolved. As the rebuttal deadline is fast approaching, we would greatly appreciate any further feedback, suggestions, or concerns at your earliest convenience. We are eager to engage in further discussion and clarify any unresolved points to ensure all matters are fully addressed.
>
> Best regards,
>
> Authors of Submission 2616

---

> ### Comment · Reviewer_mrzv · 2024-11-27
>
> Dear Authors,
>
> Thanks for your response and revisions.
>
> Your claim has addressed most of my concerns, and I believe you will implement the suggested changes regarding the concerns we discussed in the revised paper. I will raise my score from 5 to 6. However, it would be better to reflect the claim in your revised paper.

---

> > ### Author Response · Authors · 2024-11-28
> >
> > Thank you for your understanding. We have updated the PDF and highlighted our discussions in green.

---

### Official Review · Reviewer_yKyu · 2024-10-31

**Soundness:** 3
**Presentation:** 3
**Contribution:** 3
**Rating:** 8
**Confidence:** 3

**Summary:**

The paper presents a detailed explanation of the construction and evaluation of the WorkflowBench dataset, which contains many examples of workflow descriptions. The dataset is later used to fine-tune the open-source LLM Llama-3.1-8B.
The dataset creation follows a well-established and potentially reproducible approach. It starts with data gathering, expands to increase data diversity, and generates a final workflow dataset, enhancing real collected data with synthetic data.
The fine-tuning intends to overcome the limitations of existing LLMs in workflow orchestration and provide a consistent improvement in process automation by proposing an agentic approach.
The fine-tuned LLM called WorkflowLllama is detailed, and its capabilities are evaluated, showing a solid capacity to orchestrate complex workflows. Its performance was compared with commercial and open-source LLMs using the test set of Workflowbench.

**Strengths:**

The paper is well-presented and well-written overall. The authors identified a clear pain point in achieving an agentic LLM approach for process automation: lacking LLMs capable of correctly describing complex workflows. Although partially addressed by other works, as pointed out by the authors in the related work section (section 2), their approach is innovative in dealing with fairly more complex workflows than previous solutions.

Their approach to overcoming this limitation is sound and clever. They provide a fine-tuned LLM specialized to handle this kind of task.
To do so, they craft a well-defined dataset, and it is possible to highlight the process of dataset creation as one of the paper's main contributions, as crucial as the constructed dataset and the fine-tuned model.
After the data collection, the query expansion phase is especially interesting because it uses another LLM (e.g., ChatGPT) to help overcome the lack of diversity and complexity of the gathered initial data with syntactic data. The same applies to the clever use of ChatGPT in the evaluation phase.
Compared with other LLMs solving the same kind of problem, the presented results show the potential use of the author's solution to deal with the initial established problem, i.e., the automation of complex tasks.
WorkflowLlama outperforms other LLMs when the workflow demands generalization to unseen instructions and unknown APIs.

I also highlight the good approach to evaluating the effectiveness of the LLM-based evaluator (section 4.2), which strengthens the paper's arguments.

**Weaknesses:**

- The background on automation's relevance (lines 36-38) might be condensed, as automation's importance is widely recognized.
- In section 3.1, the author can better explain the $\mathcal{D}$. The other elements were presented before, but  $\mathcal{D}$ is directly introduced on line 248 without prior explanation.
- Lines 245-247 read (my emphasis):  "This bottom-up [...], efectively **ensuring content realiability** and **minimizing the risk of hallucination**". These are pretty strong assumptions without solid evidence, especially if generalized to every LLM. Do the authors know published papers that confirm them (since this paper is not about reducing hallucinations)? If not, the phrase can be removed or rephrased to be more humble.
- The paper lacks a clearer section on threats to validity. The authors provided some in Appendix E, but they should be incorporated into the main text.
- The paper also lacks a more evident running example/use case. Although the authors provide a small case study in section 4.7 and some examples in Appendix D, a consistent running example, incorporated throughout sections, could clarify the model's applications and emphasize practical use cases.

Minor issues:
- Line 52 mentions GPT-4, while the rest of the paper uses GPT-4o(-mini). Maybe it is only a typo, or the work mentioned indeed uses the GPT-4 model. Since they are different models, it can be good to double-check every reference to ensure it always talks about the correct model, maybe highlighting somewhere they are not the same.
- The size of Figure 2 can be increased to improve readability.
- Figures 3,5 and 6 could improve accessibility through higher contrast colors.
- Although well-written, the paper is sometimes repetitive (e.g., sections 1 and 3). Proofreading with this issue in mind may help the authors achieve a better final text.

**Questions:**

I am curious about the choice of ChatGPT to run both the query expansion phase of the dataset creation (section 3.2) and the experiments (sections 4.1 and 4.2). Altough I think it was a good idea, why rely on a proprietary model (the GPT-4o) with proprietary and non-disclosed prompt engineering behind it? Did the authors try to use directly the GPT-4o model? Did they try other models to run this phase? I assume that the "ease-of-use" of the ChatGPT is a fair point in favor of its use, but it should be evident in the paper.

**Details Of Ethics Concerns:**

Potential ethical considerations were taken into account in the Appendices.

---

> ### Author Response · Authors · 2024-11-22
> **Thank you for your detailed review and valuable suggestions on our paper.**
>
> **Q1:** Suggestions on paper writing, figures, formatting, layout, and test case presentation.
> **A1:** Thank you very much for your detailed review and valuable suggestions on our paper. We will carefully consider your feedback and address these issues in the revised version to make the paper clearer.

---

> ### Author Response · Authors · 2024-11-22
> **We will rephrase our statement to be more humble.**
>
> **Q2:** Lines 245-247:  "This bottom-up [...], efectively ensuring content realiability and minimizing the risk of hallucination". These are pretty strong assumptions without solid evidence, especially if generalized to every LLM. Do the authors know published papers that confirm them (since this paper is not about reducing hallucinations)? If not, the phrase can be removed or rephrased to be more humble.
>
> **A2:** Thank you very much for your detailed review. We made this claim based on our observation that directly prompting GPT-4o-mini might lead to plan generation that lacks certain critical steps. At present, we have not found research explicitly proving that the bottom-up approach significantly reduces hallucinations across all large language models. We acknowledge that this part may have been overstated, and we will revise the phrase accordingly.

---

> ### Author Response · Authors · 2024-11-22
> **Clarification on model names.**
>
> **Q3:** Line 52 mentions GPT-4, while the rest of the paper uses GPT-4o(-mini). Maybe it is only a typo, or the work mentioned indeed uses the GPT-4 model. Since they are different models, it can be good to double-check every reference to ensure it always talks about the correct model, maybe highlighting somewhere they are not the same.
>
> **A3:** Thank you for your meticulous review. In Line 52, we indeed refer to the GPT-4 model because this part discusses the challenges of orchestrating workflows as outlined in the ProAgent study [1], where they used GPT-4 as the foundation model. Additionally, we will double-check every reference in subsequent versions of the paper to improve its accuracy.
>
>
> [1] Ye, Yining, et al. "Proagent: From robotic process automation to agentic process automation." arXiv preprint arXiv:2311.10751 (2023).

---

> ### Author Response · Authors · 2024-11-22
> **We chose the GPT-4o-mini series because of its cost-effectiveness and relatively good performance.**
>
> **Q4:** I am curious about the choice of ChatGPT to run both the query expansion phase of the dataset creation (Section 3.2) and the experiments (Sections 4.1 and 4.2). Although I think it was a good idea, why rely on a proprietary model (GPT-4o) with proprietary and non-disclosed prompt engineering behind it? Did the authors try to use the GPT-4o model directly? Did they try other models for this phase? I assume that the "ease-of-use" of ChatGPT is a fair point in favor of its use, but this should be evident in the paper.
>
> **A4:** Thank you for your concern regarding the choice of foundational models in this paper.
>
> In our early experiments, we evaluated several models, including GPT-4o-mini, GPT-4o, and Claude 3.5. We ultimately selected GPT-4o-mini based on the following considerations:
>
> 1. **Performance:** GPT-4o-mini demonstrated superior performance compared to other closed-source models (e.g., Gemini Flash and Claude Haiku) on benchmarks such as MMLU, GPQA, DROP, and HumanEval. In the early experiments for this study, its performance met the requirements for both the query expansion and experimental phases.
>
> 2. **Cost Efficiency:** Given the large number of model calls required in this study, GPT-4o-mini offered a cost-effective solution, with input tokens costing 0.150 per million and output tokens costing 0.600 per million. In contrast, stronger models like GPT-4o incurred significantly higher costs, with input tokens costing 2.50 per million and output tokens costing 10.00 per million, making them less feasible within our budget constraints.
>
> We acknowledge the limitations of relying on proprietary models, particularly regarding transparency and reproducibility. To address this, we have included all prompt engineering details and data processing scripts in the supplementary material. Additionally, we remain open to exploring alternative open-source models in future research.

---

> > ### Comment · Reviewer_yKyu · 2024-11-27
> >
> > Thank you for your responses. I will keep my score and acceptance.

---

### Official Review · Reviewer_WEy1 · 2024-11-04

**Soundness:** 2
**Presentation:** 3
**Contribution:** 2
**Rating:** 6
**Confidence:** 4

**Summary:**

This paper proposes a framework that leverages LLMs to automate workflow orchestration. It extends the real world dataset by creating synthetic data through LLMs, to train the model. It conducts experiment study to compare the proposed framework with several proprietary and open source models, using CodeBLEU and Pass Rate as the metrics.

**Strengths:**

The paper is well-written and easy to follow. Examples are given to illustrate the important steps.

The extended dataset improves the diversity and complexity of the samples in the training data, allowing the models to adapt to the situations that are closer to real-world applications.

The experimental results show that the proposed framework outperformed the existing models by a large margin.

**Weaknesses:**

The paper mainly relies on prompting LLMs to generate the results of the key steps, such as data generation, commenting actions, and plan generating. It seems that there is a large space of prompt engineering to improve the performance but left to be undone.

The proposed framework bypasses the API selection when handling the OOD setting by directly serving the required APIs as the input. This greatly simplifies the problem of automating workflow orchestration.

The paper relies heavily on LLMs for the entire process, including generating the dataset, training the model, and evaluating the models (only less than 82% accuracy in a small sample with 330 instances in total). This weakens the technical contribution and the reliability of the experiment results.

**Questions:**

Are the prompts used in the framework that drive the LLMs usage automatically generated?  If yes, how to ensure the quality of the prompts? If not, how to scale up?

For each step from data generation to model evaluation, how to optimize the prompts?

The description on quality confirmation is a bit vague. The example given about the issues is not clear. How are such issues detected? How to prompt ChatGPT to refine A’ and P’, and how to ensure the quality of the refinement? How to perform the rule-based filtering, e.g., how to automatically detect if parameter constraints are violated or not?

For OOD settings, how are the other models evaluated? Do they also be given the same input like the one for Workflow Llamma, i.e., providing the list of required APIs?

Given synthetic data takes the large portion of the final benchmark (91k out of 11k) but the quality is unsure, it would be interesting to also see how Workflow Llama performs on the real-word dataset, i.e., without the synthetic data, compared to other models?

The proposed framework outperforms all the other models by an extremely large margin in the first four columns in Table 2 (such as 9.4% vs 0.5). This seems to be suspicious and needs further examination and explanation.

In Table 2, both CodeBLEU and Pass Rate are used, one examines the similarity of code and one examines if the code is executable. It would be interesting to combine both of them, such as setting a threshold on the CodeBLEU score when checking the Pass Rate to compare models, since Pass Rate itself doesn’t necessarily indicate the success of the workflow orchestration.

---

> ### Author Response · Authors · 2024-11-22
> **The manually constructed prompts have been experimentally validated as effective and are easily extendable.**
>
> **Q1**: There is a large space of prompt engineering to improve the performance but left to be undone. For each step from data generation to model evaluation, how to optimize the prompts? Are the prompts used in the framework that drive the LLMs usage automatically generated? If yes, how to ensure the quality of the prompts? If not, how to scale up?
>
> ---
>
> **A1: How prompts are crafted & More on prompt engineering:**
>
> The prompts used in our framework are not automatically generated; instead, they are manually designed and iteratively refined through experiments and manual observation to ensure their effectiveness. While we acknowledge the possibility of further improvements in prompt engineering, we emphasize that **the current prompts have already been proven effective in our experiments**.
>
> As listed in Table 2, our fine-tuned WorkflowLlama achieves **35.1%** CodeBLEU and **70.4%** pass rate under OOD settings, surpassing the strong baseline model, GPT-4o with ICL. Furthermore, WorkflowBench demonstrates strong generalization capabilities in out-of-distribution (OOD) scenarios, particularly on the T-Eval benchmark [1], where it achieves an F1 plan score of **77.5%**.
>
> ---
> **How to scale up:**
>
> We emphasize that the prompts used for dataset construction are **independent of specific tools or workflow code. This independence makes it straightforward to scale to new tools and categories**. Specifically, the process requires only the definition of general guidelines and minor adjustments of string formatting. For example, as illustrated in `Section B.5`, the query generation prompt only necessitates modifications to the ICL sample's code, the ICL sample's query, and the target query's code. This approach enables efficient query construction for every collected real-world data point.
>
> ---
> **References**
>
> [1] Chen, Zehui, et al. "T-Eval: Evaluating the tool utilization capability of large language models step by step." Proceedings of the 62nd Annual Meeting of the Association for Computational Linguistics (Volume 1: Long Papers), 2024.

---

> > ### Comment · Reviewer_WEy1 · 2024-12-02
> >
> > Since the prompts are manually generated and refined, significant human efforts would be involved in the process. This hinders the scalability of the proposed approach.
> >
> > If the paper only focuses on the testing of the effectiveness of using LLMs in workflow orchestration but overlooks the improvement of the key aspects such as prompt engineering, the paper lacks sufficient technical contribution to be accepted by a top conference like ICLR.

---

> > > ### Author Response · Authors · 2024-12-02
> > > **Clarification of Scalability and Contribution**
> > >
> > > Thank you for your feedback. Below is our response to our work's scalability and contribution:
> > >
> > > 1. We have not neglected prompt engineering. The prompts we used were manually experimented and iterated upon to ensure their effectiveness. For instance, our prompts incorporate advanced techniques such as **chain-of-thought** [1], encouraging LLMs to first develop a task plan and then generate specific workflow code.
> > >
> > > 2. The prompts we use are domain-independent (see `Appendix B`). They only require the API documentation and query to scale. **There is no need to redesign the prompt when migrating to new workflow orchestration domains in the future**.
> > >
> > > 3. The use of manually constructed prompts to extend dataset scalability is widely applied in several classic works such as Alpaca [2], ToolLLM [3], WizardLM [4], UltraFeedback [5], WizardCoder [6], MagicCoder [7], among others. These works have demonstrated that **manually designed prompts generalize well and have significantly advanced the field of LLMs**.
> > >
> > > 4. Our work goes beyond merely "testing the effectiveness of LLMs." The contribution of our work lies in collecting, transcribing, and extending raw data from platforms like Apple and RoutineHub shortcuts to propose the **first dataset aimed at enhancing workflow orchestration**. Experimental results show that our dataset effectively improves the performance of Llama-3.1-8B in both ID and OOD experiments. Notably, WorkflowLlama achieves **77.5%** performance on the OOD dataset T-Eval [8].
> > >
> > > ---
> > >
> > > ### References
> > >
> > > [1] Wei, Jason, et al. "Chain-of-thought prompting elicits reasoning in large language models." *Advances in Neural Information Processing Systems*, 35 (2022): 24824-24837.
> > >
> > > [2] Taori, Rohan, et al. *Stanford Alpaca: An Instruction-following LLaMA Model*, 2023. GitHub, https://github.com/tatsu-lab/stanford_alpaca.
> > >
> > > [3] Qin, Yujia, et al. "ToolLLM: Facilitating Large Language Models to Master 16000+ Real-world APIs." *The Twelfth International Conference on Learning Representations*.
> > >
> > > [4] Xu, Can, et al. "WizardLM: Empowering Large Pre-Trained Language Models to Follow Complex Instructions". *The Twelfth International Conference on Learning Representations*
> > >
> > > [5] Cui, Ganqu, et al. "ULTRAFEEDBACK: Boosting Language Models with Scaled AI Feedback." *Forty-first International Conference on Machine Learning*, 2024.
> > >
> > > [6] Luo, Ziyang, et al. "WizardCoder: Empowering Code Large Language Models with Evol-Instruct." *The Twelfth International Conference on Learning Representations*.
> > >
> > > [7] Wei, Yuxiang, et al. "Magicoder: Empowering code generation with oss-instruct." *Forty-first International Conference on Machine Learning*, 2024.
> > >
> > > [8]  Chen, Zehui, et al. "T-Eval: Evaluating the tool utilization capability of large language models step by step." Proceedings of the 62nd Annual Meeting of the Association for Computational Linguistics (Volume 1: Long Papers), 2024.

---

> > > > ### Comment · Reviewer_WEy1 · 2024-12-02
> > > >
> > > > Thanks for the response. I will raise my score to 6.

---

> > > > > ### Author Response · Authors · 2024-12-02
> > > > > **Thank you!**
> > > > >
> > > > > Thank you for your recognition! We greatly appreciate your suggestions and feedback.

---

> > > ### Author Response · Authors · 2024-12-02
> > > **Follow-up on Rebuttal Discussion**
> > >
> > > **Dear Reviewer WEy1**,
> > >
> > > Thank you once again for your valuable feedback.
> > >
> > > If our responses have addressed your concerns, we would be grateful if you could consider revising your rating. However, if any concerns remain, we are more than happy to continue the discussion and work on any further improvements.
> > >
> > > We truly appreciate your time and consideration, and would be grateful for any additional feedback you may have.
> > >
> > > Best regards,
> > >
> > > Authors of Submission 2616

---

> ### Author Response · Authors · 2024-11-22
> **More Details About Quality Confirmation**
>
> **Q2:** The description on quality confirmation is a bit vague. The example given about the issues is not clear. How are such issues detected? How to prompt ChatGPT to refine A’ and P’, and how to ensure the quality of the refinement? How to perform the rule-based filtering, e.g., how to automatically detect if parameter constraints are violated or not?
>
> **A2:**
>
> **How are such issues detected & how to prompt ChatGPT to refine:**
>
> We conducted a manual case study by sampling 1,000 workflows generated by the annotator model. Through this study, we identified several common issues, including:
>
> - Mismatches between task plans and the code.
> - Logical errors in Python syntax (e.g., incorrect function calls or unclosed strings).
> - Missing comments preceding each line of code.
> - The presence of meaningless binary strings.
>
> These observations were summarized and incorporated into the prompt described in `Section B.7`. We then utilized the GPT-4o-mini model to refine these workflows.
>
> ---
> **How to ensure the quality of the refinement:**
> 1. **Prompt Design:**
>    The refinement prompts were developed based on the findings from our manual case study, and their effectiveness was verified through experiments.
>
> 2. **In-Context Learning:**
>    To enhance performance, we used in-context learning by selecting examples having similar control logics (e.g., containing `if` structures or nested `if` and `for` structures) to serve as references for refinement.
>
> 3. **Model Selection:**
>    To ensure high-quality refinement, we employed OpenAI's GPT-4o-mini, a powerful proprietary model as the refinement model.
>
> 4. **Sampling Inspection:**
> We conducted manual sampling to ensure the quality of the refinement. We found that **94.2%** of the samples showed improvements in at least one of areas such as Python syntax, code naming and prompt clarity, and the correspondence between the code and queries compared to the pre-refinement version (see case study 1 below).
>
> ---
> **How to automatically detect if parameter constraints are violated:**
>
> We utilized the Python interpreter to automatically detect syntax errors. Specifically, we constructed test functions consistent with the API definitions and executed the program. If the Python program produced an error during execution, we considered it a violation of the parameter constraints and discarded the corresponding sample.

---

> ### Author Response · Authors · 2024-11-22
> **Case Study 1**
>
> **Query:**
> How can I create an engaging and interactive digital storybook for children that allows them to explore different themes and characters? The storybook should include features such as resizing images for better fit on various devices, ...(more queries)
>
> **Pre-refinement Code:**
> ```python
> vpn_measurement = is_workflow_actions_measurement_create( WFMeasurementUnit={"Unit": b/s, "Magnitude": 1}, WFMeasurementUnitType='''Data Rate''')
>
> ...(more code)
>
> created_folder = is_workflow_actions_dropbox_createfolder( WFFilePath=f'''f\'{input("Please enter the value:")}\'''', WFFileStorageService='''Dropbox''')
>
> ...(more code)
>
> saved_document = is_workflow_actions_documentpicker_save( WFFolder=folder_in_icloud_drive, WFInput=if_result, WFAskWhereToSave=False)
> ```
>
> **After-refinement Code:**
> ```python
> vpn_measurement = is_workflow_actions_measurement_create(
>     WFMeasurementUnit={'Unit': 'b/s', 'Magnitude': 1},
>     WFMeasurementUnitType='Data Rate'
> )
>
> ...(more code)
>
> folder_name = input('Please enter a folder name for your story project: ')
>
> created_folder = is_workflow_actions_dropbox_createfolder(WFFilePath=folder_name)
>
> ...(more code)
>
> image_file = input('Please enter the image file path to resize: ')
>
> resized_image = is_workflow_actions_image_resize(image_file, dimensions='fit')
>
> ...(more code)
> ```
>
> **Comparison**
> 1. **Python Syntax**:
> The pre-refinement code contains ambiguous constructs and syntax inconsistencies, such as unquoted strings (`b/s` instead of `'b/s'`) and improperly formatted string interpolation (e.g., `f'''f\'{input("Please enter the value:")}\''''`). In the after-refinement version, these issues are resolved by adhering to proper Python syntax, including quoting strings appropriately, using well-structured functions, and ensuring correct syntax for function calls and inputs. This improves both readability and functionality.
>
> 2. **Correspondence Between Code and Queries**:
> The pre-refinement code fails to align fully with the user's requirements, omitting critical functionality like resizing images for better device compatibility. The after-refinement version explicitly addresses these gaps by adding functionality such as `is_workflow_actions_image_resize` for resizing images.

---

> ### Author Response · Authors · 2024-11-22
> **WorkflowLlama performs better with synthetic data on on the real-world dataset.**
>
> **Q3**: Given synthetic data takes a large portion of the final benchmark (91k out of 111k), but its quality is uncertain, it would be interesting to see how Workflow Llama performs on the real-world dataset, i.e., without the synthetic data, compared to other models?
>
> **A3**: We appreciate your question regarding the evaluation. It should be clarified that to ensure the reliability of the evaluation, **all test samples are collected from real-world data**.
>
> **Synthetic data is used exclusively for training phase to enhance the model**. To verify the impact of the synthetic data, we have conducted the ablation study in the `Section 4.6 Ablation Study` (`Table 4`). Removing synthetic data during training results in consistent declines across the CodeBLEU metric and its four subcomponents. This demonstrates the effectiveness of incorporating synthetic data .

---

> ### Author Response · Authors · 2024-11-22
> **BLEU and weighted N-gram are sensitive to the implementation details. So we also employ the pass rate metric.**
>
> **Q4**: The proposed framework outperforms all the other models by an extremely large margin in the first four columns in Table 2 (such as 9.4% vs 0.5). This seems to be suspicious and needs further examination and explanation.
>
> **A4**: We appreciate the reviewer’s observation regarding the significant performance differences in the first four columns of Table 2. Below, we provide a detailed explanation to address this concern, including a deeper dive into the metrics used and additional case studies.
>
>  ---
> **Metrics Interpretation and Their Limitations**
>
> Both **BLEU** and **weighted N-gram** metrics measure the **degree of overlap** between candidate and reference texts, primarily focusing on surface-level features like n-gram matches:
>
> 1. **BLEU**: Calculates modified precision by assessing the proportion of n-grams in the candidate that also appear in the reference text.
> 2. **weighted N-gram**: Extends this by assigning weights to specific n-grams, emphasizing critical components (e.g., syntax or API calls) in the evaluation.
>
>
> Given these properties, these metrics are particularly **sensitive to the implementation details of candidate code** (see case study 2 below), such as whether custom functions are defined, the use of `match...case... ` structures, or the inclusion of a `main` function. WorkflowLlama, trained on datasets similar to the test sets in the solution implementation, produces outputs closely aligned with reference texts in these aspects. Consequently, it achieves higher BLEU and weighted N-gram scores, especially when compared to models that generate outputs with greater variability.
>
> **To address the potential bias of these metrics toward surface-level similarity, we also employ the pass rate metric**, which evaluates the functional correctness of generated code. This ensures that models are assessed on their ability to complete the intended tasks, irrespective of formatting alignment (see case study 2 below).
>
>
>
> ---
> ### Case Study 2
>
> **Query:**
> What steps would I need to follow to develop a script that allows a user to select specific YouTube videos corresponding to various amusement park rides?
>
> **Reference Code (GPTEval: True):**
> ```python
> # Initiates pattern matching on the function that fetches user input.
> match input():
>     # Defines a case for when the input is 'Blast Off'.
>     case "Blast Off":
>         # Sets the 'blast_off_url' variable to a specific YouTube video URL when input matches 'Blast Off'.
>         blast_off_url = is_workflow_actions_openurl( Show-WFInput=True, WFInput='''https://www.youtube.com/watch?v=nKOtGJECa_c''')
>     # Defines a case for when the input is 'Boomerang'.
>     case "Boomerang":
>         # Sets the 'boomerang_url' variable to a specific URL when input matches 'Boomerang'.
>         boomerang_url = is_workflow_actions_openurl( Show-WFInput=True, WFInput='''https://www.youtube.com/watch?v=bTK0ymzGV4g''')
>     # Defines a case for when the input is 'Mind Eraser'.
>     case "Mind Eraser":
>         # Sets the 'mind_eraser_url' variable to a URL when input matches 'Mind Eraser'.
>         mind_eraser_url = is_workflow_actions_openurl( Show-WFInput=True, WFInput='''https://www.youtube.com/watch?v=VXW_8N1Q-TM''')
> ...(more code)
> ```
>
> **Candidate Code by GPT-4o (CodeBLEU: 11%, BLEU: 0.2%, weighted N-gram: 0.2%, AST: 16.1%, dataflow:13.1%, GPTEval: True):**
> ```python
> # Prompt the user to enter the amusement park ride name.
> Input = f'{input("Please enter the amusement park ride: ")}'
>
> # Construct the YouTube search URL by appending the user input to the base URL.
> search_url = f'https://www.youtube.com/results?search_query={Input}'
>
> # Open the constructed YouTube search URL in Safari.
> is_workflow_actions_openurl(WFInput=search_url)
> ```
>
> **Analysis:**
>
> The reference code uses `match-case` to explicitly map rides to specific YouTube links, while the Candidate Code dynamically generates a search link based on user input. Although both fulfill the task and evaluated `True` by GPT4o-mini, the Candidate Code’s simpler approach results in a very low BLEU and weighted N-gram score due to different code structures.
>
>
> ---

---

> ### Author Response · Authors · 2024-11-22
> **It is unnecessary for Pass Rate to depend on CodeBLEU.**
>
> **Q5:** In Table 2, both CodeBLEU and Pass Rate are used, one examines the similarity of code and one examines if the code is executable. It would be interesting to combine both of them, such as setting a threshold on the CodeBLEU score when checking the Pass Rate to compare models, since Pass Rate itself doesn’t necessarily indicate the success of the workflow orchestration.
>
> **A5:**
> Thank you for your suggestions regarding the evaluation metrics of this paper.
> We would like to clarify that, as mentioned in `Appendix B.2` and `Rebuttal A4`, **Pass Rate is not a metric for determining whether the code is executable but rather a measure of the functional correctness of the code, i.e., indicating the success of the workflow orchestrate indeed**. Therefore, we believe that it is unnecessary for Pass Rate to depend on CodeBLEU and it can be calculated independently to evaluate the correctness of the workflow. We will emphasize its calculation logic and its relationship with CodeBLEU in the revised version.

---

> ### Author Response · Authors · 2024-11-24
> **For OOD settings, all models are evaluated consistently by providing a unified list of required APIs.**
>
> **Q6**: For OOD settings, how are the other models evaluated? Do they also be given the same input like the one for Workflow Llamma, i.e., providing the list of required APIs?
>
> **A6**: **To ensure experimental fairness**, as shown in Figure 3, the evaluation of baselines under OOD settings is conducted in the same manner as WorkflowLlama. All models are directly provided with the required APIs as input to orchestrate workflows based on the user's query.

---

> ### Author Response · Authors · 2024-11-26
> **Experimental results show that WorkflowLlama remains effective in handling OOD settings, where golden APIs are not available during inference.**
>
> **Q7**: The proposed framework bypasses the API selection when handling the OOD setting by directly serving the required APIs as the input. This greatly simplifies the problem of automating workflow orchestration.
>
> **A7:** Thank you for your insightful comments on our experimental setup. The main goal of our original OOD experiment was to show that WorkflowLlama can generalize effectively even when it encounters APIs that were not seen during training. Our main concern was introducing a retrieval mechanism can be problematic, as the retriever may select irrelevant APIs, which could negatively impact performance. This makes it challenging to determine whether any performance changes are due to the retriever or the model's ability to generate workflows.
>
> In addition, following your suggestion, we conducted **additional experiments under a setting where golden APIs are not used**. The results demonstrate that **in OOD scenarios where golden APIs are not provided, WorkflowLlama outperforms GPT-4o with ICL, proving the effectiveness of our framework**.
>
> The specific experimental setup and results are as follows:
>
> ---
>
> In the workflow orchestration task, a single query may correspond to dozens of tools. Selecting the appropriate APIs for a given query requires advanced reasoning capabilities, where a simple dual-tower model for semantic matching [1] may fail. Therefore, we prompt large language models with ICL samples to extract APIs. The specific prompt is as follows:
>
> ```
> You are an API retriever who can extract the APIs needed to complete the query based on the user's query and an API list.
> API list:
> {APIs with description}
> Here are some examples:
> {examples}
> So for this query, which APIs are needed to complete this task? Please return a list of required apis without any explanation.
> query: {query}  APIs:
> ```
> where we select samples with the most similar queries in the training set using the `all-MiniLM-L6-v2` model.
>
>
> The experimental results for this API retriever are as follows:
> | LLMs                         | Precision | Recall  |
> |------------------------------|-----------|---------|
> | GPT-4o-mini                  | **42.5%** | 36.4%   |
> | Qwen2.5-72B                  | 40.6%     | **40.7%**|
>
> Although the retriever's performance metrics are not particularly high, manual inspection of 50 randomly selected cases using semantic matching revealed that 84% of the retrieved APIs successfully fulfilled the tasks specified in the queries. To further validate these findings, we **trained WorkflowLlama with golden APIs** and conducted workflow orchestration experiments **using the retrieved APIs in an OOD setting**. The results are as follows:
>
> | APIs source          | Pass Rate |
> |---------------------|-----------|
> | golden              | **70.4%** |
> | GPT-4o-mini         | 66.7%     |
> | Qwen2.5-72B         | 69.0%     |
>
> Based on the results, we observe that **WorkflowLlama, trained with golden APIs, shows only a minor performance decrease when utilizing APIs retrieved by LLMs**, yet it still outperforms powerful closed-source models such as GPT-4o with ICL. Moreover, when using APIs retrieved by the open-source model Qwen2.5, WorkflowLlama achieves even better results than GPT-4o-mini.
>
>
> Therefore, we want to emphasize that the main experimental setup of this paper does not greatly simplify the problem of automating workflow orchestration. Instead, it demonstrates that the framework remains effective even in the absence of golden APIs during inference.
>
> ---
> **References**
>
> [1] Qin, Yujia, et al. "ToolLLM: Facilitating Large Language Models to Master 16000+ Real-world APIs." The Twelfth International Conference on Learning Representations. 2023.

---

> ### Author Response · Authors · 2024-11-26
> **Clarification of Evaluation Metrics and Validation in Our Study**
>
> **Q8**: The paper relies heavily on LLMs for evaluating (only less than 82% accuracy in a small sample with 330 instances in total). This weakens the technical contribution and the reliability of the experiment results.
>
>
> **A8**: Thank you for your valuable feedback regarding the evaluation process in our paper. Below is our response to the concern:
>
> **LLMs-based Evaluation as a Common Practice**: Using large language models (LLMs) for automatic evaluation is a widely accepted practice in the field of instruction-tuned LLMs [1,2,3,4,5]. This approach is not unique to our research but is commonly adopted due to its efficiency and reasonable accuracy in providing quick assessments of model performance. The use of ChatGPT for evaluation in our work is driven by the need for scalability and efficiency in the evaluation process.
>
>
> **Additional Human-annotated Results**: To further validate the reliability of our experimental results, we manually annotated **all 1190 test instances** of WorkflowLlama and GPT-4o with In-Context Learning (ICL). The experimental results are as follows:
> | Model                     | Human Pass Rate       |
> |---------------------------|-----------------------|
> | GPT-4o with ICL           | 65.9%                 |
> | WorkflowLlama              | **71.1%**             |
>
> From these results, it is evident that, even with fully human evaluation, WorkflowLlama outperforms the strong baseline model, GPT-4o with ICL.
>
>
> **Other Evaluation Metrics Employed**: It should be noted that we did not rely solely on LLMs-based evaluation in this paper. For the results on WorkflowBench, we also used the CodeBlEU scores to ensure the objectivity of the evaluation. For out-of-distribution generalization experiments on T-Eval [6], we calculated the plan scores using ground truth data. The promising results demonstrate the strong capabilities of WorkflowLlama.
>
> ---
>
> **References**
>
> [1] Liu, Yang, et al. "Gpteval: Nlg evaluation using gpt-4 with better human alignment." arXiv preprint arXiv:2303.16634 (2023).
>
> [2] Dubois, Yann, et al. "Alpacafarm: A simulation framework for methods that learn from human feedback." arXiv preprint arXiv:2305.14387 (2023).
>
> [3] Chiang, Wei-Lin, et al. "Vicuna: An open-source chatbot impressing gpt-4 with 90%* chatgpt quality." See https://vicuna. lmsys. org (accessed 14 April 2023) (2023).
>
> [4] Chiang, Cheng-Han, and Hung-yi Lee. "Can Large Language Models Be an Alternative to Human Evaluations?." arXiv preprint arXiv:2305.01937 (2023).
>
> [5] Qin, Yujia, et al. "ToolLLM: Facilitating Large Language Models to Master 16000+ Real-world APIs." The Twelfth International Conference on Learning Representations. 2023.
>
> [6] Chen, Zehui, et al. "T-eval: Evaluating the tool utilization capability of large language models step by step." Proceedings of the 62nd Annual Meeting of the Association for Computational Linguistics (Volume 1: Long Papers). 2024.

---

> ### Author Response · Authors · 2024-11-26
> **Follow-up on Rebuttal Discussion and Further Feedback**
>
> Dear Reviewer WEy1,
>
> I hope this message finds you well. We appreciate the time and effort you have put into reviewing our paper and providing valuable feedback. As the rebuttal deadline is fast approaching, we would like to kindly follow up and see if there are any remaining points or concerns that we could address.
>
> We are eager to engage in further discussion and would greatly appreciate any additional thoughts or suggestions you might have. Please do not hesitate to share any further feedback, and we will ensure a prompt response.
>
> Thank you once again for your time and consideration.
>
> Best regards,
>
> Authors of Submission 2616

---

> ### Author Response · Authors · 2024-11-27
> **Kind Request for Your Feedback Before the Rebuttal Deadline**
>
> Dear Reviewer WEy1,
>
> We have diligently addressed all the concerns raised during the rebuttal period, and we would like to kindly ask if any remaining issues have been resolved. As the rebuttal deadline is fast approaching, we would greatly appreciate any further feedback, suggestions, or concerns at your earliest convenience. We are eager to engage in further discussion and clarify any unresolved points to ensure all matters are fully addressed.
>
> Best regards,
>
> Authors of Submission 2616

---

> ### Author Response · Authors · 2024-11-28
> **Looking for Further Discussion and Feedback**
>
> Dear Reviewer WEy1,
>
> We hope this message finds you well. We would like to inform you that we have updated our manuscript and highlighted the revisions made in response to your valuable comments in **red**. As we have been awaiting your feedback for nearly five days, we kindly request your review of the rebuttal and the revised PDF at your earliest convenience. Should there be any points we have not addressed sufficiently, we would be grateful for the opportunity to discuss them further.
>
> Thank you for your time and consideration.
>
> Best regards,
>
> Authors of Submission 2616

---

> ### Author Response · Authors · 2024-11-29
> **Thanks to Reviewer WEy1**
>
> Dear Reviewer WEy1,
>
> Thank you for your valuable feedback and contributions to improving our work. We have made revisions in the updated version, with changes highlighted in red. If you feel that your concerns have been adequately considered, we kindly ask that you consider raising your rating. However, if you still have any remaining questions or reservations, we would be happy to discuss them further and provide any clarifications.
>
> Best regards,
>
> The Authors of Submission 2616

---

> ### Author Response · Authors · 2024-11-30
> **Follow-up on Rebuttal Discussion**
>
> Dear Reviewer WEy1,
>
> We hope you’re enjoying a wonderful Thanksgiving and a relaxing weekend.
>
> Thank you again for your valuable feedback. If your concerns have been addressed, we would appreciate it if you could consider updating your rating. If there are any remaining issues, we are happy to continue the discussion.
>
> If you have time, we would be grateful for your response.
>
> Best regards,
>
> Authors of Submission 2616

---

> ### Author Response · Authors · 2024-12-02
> **Follow-up on Discussion**
>
> **Dear Reviewer WEy1,**
>
> Thank you for your thoughtful review and feedback. Given that the discussion period is now less than **2 days**, we kindly ask for your prompt engagement or feedback on our submission so we can finalize the review process.
>
> Your timely response would be greatly appreciated.
>
> Best regards,
>
> Authors of Submission 2616

---

### Official Review · Reviewer_x9fK · 2024-11-11

**Soundness:** 2
**Presentation:** 1
**Contribution:** 2
**Rating:** 5
**Confidence:** 4

**Summary:**

The paper argues that state-of-the-art models like GPT-4o face challenges in effectively handling complex workflows. To address this, the paper introduces WorkflowLLM, a data-centric framework designed to enhance the workflow orchestration capabilities of LLMs. Central to this framework is WorkflowBench, a large-scale fine-tuning dataset comprising 106,763 workflows, 1,503 APIs, and 83 applications across 28 categories. WorkflowBench is constructed through a three-phase pipeline: data collection from sources like RoutineHub, query expansion using ChatGPT to create diverse and complex workflows, and workflow generation leveraging a trained annotator model for high-quality workflow synthesis. The dataset is enriched with Python-style code, comments, and hierarchical thought structures to improve LLM learning.

Based on WorkflowBench, the paper fine-tunes Llama-3.1-8B, resulting in WorkflowLlama, which demonstrates superior performance compared to existing models, including GPT-4o, on workflow orchestration tasks. WorkflowLlama achieves notable success in generalization to unseen APIs and tasks, evaluated using metrics such as CodeBLEU and Pass Rate. Additionally, WorkflowBench exhibits robust zero-shot generalization on the T-Eval benchmark, achieving an F1 plan score of 77.5%.

**Strengths:**

S1)The observation and problems are interesting and relevant to the conference

S2) The dataset might have potential uses

S3)The experiments involve many systems

**Weaknesses:**

W1) The scientific or technical contributions are limited as the key contributions of paper are around data curation effort. And the significance of data size and quality  is not clear (see Q2).

W2) Section 3 mentions many steps to generate the needed data for its workflow using ChatGPT but the Quality control protocol is not very clear. Even the author provides some examples and algorithms in appendix, the details are very descriptive . Also section4.2  mentions using  human evaluator to re-label the sampled data,It’d be better to provide more details towards the quality control protocol for this human-driven process as well.

W3) Many technical details are not very clear (see questions)

W4) Code is provided in supplemental material but there is no document how to reproduce the experiment results ( also see Q3)

**Questions:**

Q1)The “pass rate” is an important metric to evaluate the performance in section 4.3. Could you elaborate more on how it is calculated? What are the reasons to choose it? And how is related to training/fine-tuning phases?

Q2)Could you  contextualize the size and the quality of the generated dataset   in terms of  how significant it is in comparison with SOTA or related works? Are they general enough for RPA for just for applications around Apple Shortcuts and RoutineHub?

Q3) The papers also mention about fine-tuning Worfflow Lalama and annotator in a very short paragraph in section. Could you elaborate more about setup, why such training parameters are chosen?

---

> ### Author Response · Authors · 2024-11-22
> **Technical contributions & Generalizability of WorkflowBench**
>
> **W1 & Q2 (Part 1)**: The scientific or technical contributions are limited as the key contributions of paper are around data curation effort. Are the dataset general enough for RPA for just for applications around Apple Shortcuts and RoutineHub?
>
> **Response**:
>
> 1. **Importance of Workflow Automation and Contribution of Our WorkflowBench**
>
> The goal of workflow automation is to **automate repetitive tasks to minimize human labor and improve efficiency**, a widely adopted practice in modern life. For instance:
>
>     -  Over 2.2 million businesses globally use the Zapier platform to automate office tasks [1].
>     -  More than 2 million developers leverage the UiPath platform for creating automation tools [2].
>
> Despite the success of advanced platforms like UiPath, these tools rely heavily on manual efforts to construct workflows by dragging and dropping components. This remains labor-intensive and limits scalability. Recent advancements in large language models (LLMs) promise automated workflow generation. However, the capability of current LLMs, including advanced GPT-4, to orchestrate workflows is limited—they can only handle an average of 6.1 actions [3] while Apple Shortcuts involves an average of 70.4 actions, insufficient for real-world applications.
>
> To address this limitation, our work **takes the lead to introduce WorkflowBench**, a large-scale, diverse, and sufficiently complex workflow orchestration dataset. This dataset aims to empower LLMs to effectively construct workflows, addressing a critical gap in current automation capabilities.
>
> 2. **Generalizability Beyond Apple Shortcuts and RoutineHub**
>
> Although our dataset is derived from Apple Shortcuts and RoutineHub, our data processing ensures it possesses generalization capabilities that extend beyond these platforms. Our experiments on T-Eval further support this claim.
>
>  **Format Conversion**：Apple Shortcuts uses the property list (plist) format, which is deeply embedded in the macOS and iOS ecosystems. This format: (1) lacks portability to other platforms. (2) is challenging to interpret due to its use of UUIDs and hexadecimal group identifiers for variables and control logic (see Appendix C for a case study). To overcome these limitations, we adopt a generalized Python format to represent function interfaces and invocation processes (see Algorithm 1 for the transcrbing algorithm). This universal representation enhances dataset generalizability beyond MacOS and iOS platforms, streamlines LLMs' training by abstracting workflows into a flexible and harmonized structure.
>
> **Data Expansion**：We found that real-world workflows collected from RoutineHub and similar platforms exhibited: (1) skewed workflow categories, concentrated in utility, productivity, and games domains (see Figures 4.b for detailed statistics) and (2) the lack of use of third-party apps (see Figures 4.c for detailed statistics). To address this imbalance, we first (1) expanded the dataset by generating workflows for a broader range of queries and then (2) improved category diversity and incorporated a wider variety of APIs (see Figures 4.e and 4.f for detailed statistics).
>
> As shown in **Section 4.5**, WorkflowBench demonstrates strong generalization capabilities in OOD scenarios. Specifically, on the T-Eval benchmark [4], it achieves an **F1 plan score of 77.5%**, showcasing its utility and generalization.
>
> ---
>
> **References**
>
> [1] [Zapier Community](https://community.zapier.com/)
> [2] [UiPath: About Us](https://www.uipath.com/about-us)
> [3] Ye, Yining, et al. "Proagent: From robotic process automation to agentic process automation." arXiv preprint arXiv:2311.10751 (2023).
>
> [4] Chen, Zehui, et al. "T-Eval: Evaluating the tool utilization capability of large language models step by step." Proceedings of the 62nd Annual Meeting of the Association for Computational Linguistics (Volume 1: Long Papers), 2024.

---

> ### Author Response · Authors · 2024-11-22
> **Comparison with SOTA or related works**
>
> **W1 & Q2 (Part 2)**: The significance of data size and quality is not clear. Could you contextualize the size and the quality of the generated dataset in terms of how significant it is in comparison with SOTA or related works?
>
> **Response**:
>
> We would like to emphasize that **the dataset introduced in this work is fundamentally distinct from prior datasets**. To the best of our knowledge, **WorkflowBench is the first dataset explicitly designed for workflow generation, emphasizing the creation of reusable code leveraging predefined tools**. While related datasets such as ToolBench [1], API-Bank [2], and APIBench [3] involve tool usage, their primary focus is on dynamically performing API selection and parameter filling based on a user-specified task requirement.
>
> Moreover, it is challenging to adapt these datasets for workflow orchestration tasks. Specifically, ToolBench [1], API-Bank [2], and APIBench [3] only contain average action counts of 4.0, 2.1, and 1.0, respectively. Additionally, they are all linear in structure, lacking logical control structures such as `if` and `for`.
> In contrast, WorkflowBench is specifically designed for generating multi-step workflows, enabling the automation of complex and repetitive tasks. Specifically, **WorkflowBench contain an average of more than 70 actions** (see Table 1 for detailed statistics), and **includes an average of 7.9 `if` logic statements and 0.5 `for` loops**.
>
> Therefore, WorkflowBench is a novel dataset that is significantly more complex than existing works.
> We hope this dataset will inspire researchers to address more complex and realistic tasks.
>
> ---
> **References**
>
> [1] Qin, Yujia, et al. "ToolLLM: Facilitating Large Language Models to Master 16000+ Real-world APIs." The Twelfth International Conference on Learning Representations.
>
> [2] Li, Minghao, et al. "API-Bank: A Comprehensive Benchmark for Tool-Augmented LLMs." Proceedings of the 2023 Conference on Empirical Methods in Natural Language Processing. 2023.
>
> [3] Patil, Shishir G., et al. "Gorilla: Large language model connected with massive apis." arXiv preprint arXiv:2305.15334 (2023).

---

> ### Author Response · Authors · 2024-11-22
> **The pass rate, evaluated using GPT-4o-mini, is a metric for assessing  the functional correctness and is independent of the training process.**
>
> **Q1**: How is Pass Rate Evaluated, Why is it Used, and How is it Related to Training?
>
> **Response**:
>
> ---
>
> **How pass rate is calculated**:
>
> We evaluate the pass rate using `GPT-4o-mini` as the assessment model. **The pass rate represents the proportion of functionally correct workflow code using only the given APIs.**
>
> Specifically, for each sample to be evaluated, we construct a prompt that includes the sample's query, API documentation, and the workflow code in the following format (as also described in `Section B.2`):
> ```
> You are a kindly code reviewer, I will provide you with a query, a list of allowed apis and a piece of code to be reviewed, you help me to check if the code to be reviewed is compliant with our specifications.
> The requirements are as follows:
> 1. You **should return True even if the code implements additional functionality not required in the query**, as long as it roughly implements the requirements in the query.
> 2. We don't impose any requirements on code readability or naming conventions. You **should return True as long as the reviewed code doesn't use disallowed functions and reasonably accomplishes what is asked in the query in general terms**. There's no need to get strictly hung up on the details.
> 3. Return False if the code fails to fulfill the requirement in the query. e.g. if it is proposed in the query to turn down the battery level of the phone and the brightness of the screen, it is a failure to fulfill only any one of the functions.
> 4. Built-in python syntax such as `if`, `loop`, `input()`, and `print()` are allowed.  Return False if the code uses **any external functions or apis** not in allowed apis list and not a built-in function such as input(), print(). For example, if I provide the is_workflow_openurl function, this should be used. Any use of any other library like requests etc. is a False.
> query:{query}
> list of allowed apis: {apis}
> code to review: {code}
>
> Your answer: [True or False with interpretation]
> ```
> This prompt is fed into GPT-4o-mini, which produces a binary classification (`True` or `False`). For each method (e.g., GPT-4o or WorkflowLLama), we calculate the pass rate as the proportion of samples classified as `True` out of the total number of samples (`True + False`).
>
> ---
>
> **Why choose pass rate**:
>
> The another used metric CodeBLEU is a reference-based evaluation metric that assesses the similarity between candidate and reference code by analyzing text overlap, syntactic structure, and semantic data flow. However, similarity to reference code does not guarantee functional correctness.
> Solving a problem often allows for highly diverse code implementations (see case study below). For this reason, we use the pass rate metric, which directly **measures whether the generated code meets the requirements specified in the query**, independent of its similarity to reference implementations.
>
> ---
>
> **Relation to training/fine-tuning phases**:
>
> The evaluation process using `GPT-4o-mini` is **entirely independent of the training phase**. We did not introduce the evaluate model during the training/fine-tuning process to ensure our model not to overfit the evaluation metric.

---

> > ### Author Response · Authors · 2024-11-22
> > **Case Study**
> >
> > **Query:**
> >
> > What steps would I need to follow to develop a script that allows a user to select specific YouTube videos corresponding to various amusement park rides?
> >
> > **Reference Code:**
> > ```python
> > # Initiates pattern matching on the function that fetches user input.
> > match input():
> >     # Defines a case for when the input is 'Blast Off'.
> >     case "Blast Off":
> >         # Sets the 'blast_off_url' variable to a specific YouTube video URL when input matches 'Blast Off'.
> >         blast_off_url = is_workflow_actions_openurl( Show-WFInput=True, WFInput='''https://www.youtube.com/watch?v=nKOtGJECa_c''')
> >     # Defines a case for when the input is 'Boomerang'.
> >     case "Boomerang":
> >         # Sets the 'boomerang_url' variable to a specific URL when input matches 'Boomerang'.
> >         boomerang_url = is_workflow_actions_openurl( Show-WFInput=True, WFInput='''https://www.youtube.com/watch?v=bTK0ymzGV4g''')
> >     # Defines a case for when the input is 'Mind Eraser'.
> >     case "Mind Eraser":
> >         # Sets the 'mind_eraser_url' variable to a URL when input matches 'Mind Eraser'.
> >         mind_eraser_url = is_workflow_actions_openurl( Show-WFInput=True, WFInput='''https://www.youtube.com/watch?v=VXW_8N1Q-TM''')
> > ...(more code)
> > ```
> >
> > **Candidate Code (CodeBLEU: 11%, BLEU: 0.2%, weighted N-gram: 0.2%, AST: 16.1%, dataflow:13.1%, GPTEval: True):**
> > ```python
> > # Prompt the user to enter the amusement park ride name.
> > Input = f'{input("Please enter the amusement park ride: ")}'
> >
> > # Construct the YouTube search URL by appending the user input to the base URL.
> > search_url = f'https://www.youtube.com/results?search_query={Input}'
> >
> > # Open the constructed YouTube search URL in Safari.
> > is_workflow_actions_openurl(WFInput=search_url)
> > ```
> >
> > **Analysis:**
> >
> > In this case study, the candidate code has a relatively low CodeBLEU score (only 11%). However, the candidate code effectively achieves the same functionality by dynamically generating YouTube search URLs based on user input, offering flexibility and scalability over hardcoded mappings. Therefore, we believe that solely using the CodeBLEU metric is insufficient to fully evaluate the correctness of the workflow, so we introduced the pass rate metric.

---

> ### Author Response · Authors · 2024-11-22
> **Details of Quality Control Protocol and Human Evaluation Criteria.**
>
> **W2**: Details of Quality Control Protocol.
>
> **Response**:
>
> Our quality control process involves the following three steps:
>
> 1. **Workflow Refinement Using GPT-4o-mini with In-Context Learning:** We utilize GPT-4o-mini to refine workflows generated by the annotator. Specifically, we focus on minimizing Python syntax errors, ensuring every line of code is preceded by an explanatory comment, and reducing meaningless binary encoding strings. For detailed prompts used in this process, please refer to `Section B.7`. We conducted manual sampling to ensure the quality of the refinement. We found that **94.2%** of the samples showed improvements in at least one of areas such as Python syntax, code naming and prompt clarity, and the correspondence between the code and queries compared to the pre-refinement version.
>
> 2. **Removal of Empty Code Outputs:** We observed that GPT-4o-mini occasionally generates empty code outputs when the input or output task plan is too lengthy. These instances were identified and excluded from our dataset. As a result, approximately 3% of the samples, about **3,107**, were filtered out at this stage.
>
> 3. **Filtering for Function and API Consistency:**
>
>     We utilized the Python interpreter to automatically detect syntax errors. Specifically, we constructed test functions consistent with the API definitions and executed the program. If the Python program produced an error during execution, we considered it a violation of the parameter constraints and discarded the corresponding sample. For instance, for the following function:
>
> ```python
>     def is_workflow_actions_openurl(WFInput: str) -> None:
> ```
>
> Samples that either failed to pass the WFInput parameter or passed more than the required WFInput parameter were excluded. In this step, we filtered out **19,506** samples, which account for approximately 18% of the synthetic data.
>
> ---
>
> **W2**: Details of Human Evaluation Criteria.
>
> **Response**:
>
> Thank you for elaborating on our human evaluation process, which is used to verify the reliability of the GPT-4o-mini-based evaluation for calculating the pass rate.
> The human evaluation criteria used in this work are **fully aligned** with the prompt provided to GPT-4o-mini for computing the pass rate, which can be found in `Section B.2`. Specifically, the evaluation assesses **whether the workflow code fulfills the requirement using only the functions specified in the API documentation, without imposing requirements on code readability or naming conventions**.

---

> ### Author Response · Authors · 2024-11-22
> **More Experimental Setup and Selection Reasons.**
>
> **Q3**: Could you elaborate more about the setup and explain why these training parameters were chosen?
>
> **Response:**
>
> 1. **Experimental Setup**:
>
> We have supplemented the experimental setup described in Section 4.1 as follows:
>
> | **Hyperparameter**         | **Value**                   |
> |----------------------------|-----------------------------|
> | Context Length             | 8192                        |
> | Epochs                     | 3                           |
> | Hardware                   | 8 GPUs (40GB A100 each)     |
> | Batch Size                 | 2 per device                |
> | Gradient Accumulation Steps| 2                           |
> | Precision                  | torch.bfloat16              |
> | Warmup Ratio               | 0.1                         |
> | LR Scheduler               | Linear                      |
> | Learning Rate              | 2e-5                        |
> | Memory Optimization        | Deepspeed Zero3             |
>
> 2. **Reasoning Behind Choices**:
>    - **Model Selection**: We conducted a preliminary experiment by feeding query, API documentations and golden task plan (which is different from the settings in the paper) as input on the real-world data. We performed LoRA fine-tuning for 5000 steps, finding that Llama3-8B achieved a BLEU score of 9.5%, while Llama3.1-8B achieved 11.6%. Therefore, we selected Llama3.1-8B as the base model for subsequent experiments.
>
>    - **Finetuning Strategy**: Similar to model selection, we observed that full fine-tuning of Llama3.1-8B, under identical experimental settings as in the model section part, achieved a BLEU score of 14.4%. Thus, we chose full fine-tuning for subsequent experiments.
>
>    - **Learning Rate**: We experimented with various learning rates, including 1e-5 and 5e-5, for fully fine-tuning Llama3.1-8B. Among these, we found that a learning rate of 2e-5 yielded the best performance. The detailed results are summarized below.
>
> | Learning Rate  | Average BLEU      |
> |----------------|-------------------|
> | 1e-5           | 14.3%             |
> | 2e-5           | 14.4%             |
> | 5e-5           | 13.9%             |
>
>    - **Epochs**: Under the experimental setting as in the paper, we found that the model's CodeBLEU score converged on the development set after 3 epochs. Therefore, we chose to fine-tune for 3 epochs. The experimental results are as follows:
>
> | Epoch           | Average CodeBLEU |
> |-----------------|------------------|
> | Before Training | 0.244            |
> | Epoch 1         | 0.328            |
> | Epoch 2         | 0.375            |
> | Epoch 3         | 0.393            |
> | Epoch 4         | 0.394            |
>
>    - **Context Length**:
>   A length of 8192 was chosen as it covers over 95% of the samples in the dataset. The following table shows the token length percentiles of samples in the WorkflowBench.
>
> | **Percentile (%)** | **Sample Length** |
> |--------------------|-------------------|
> | 50                 | 2754.0            |
> | 60                 | 3116.0            |
> | 70                 | 3547.0            |
> | 80                 | 4109.0            |
> | 90                 | 5067.0            |
> | 95                 | 6124.4            |
> | 99                 | 11098.8           |
>
>    - **Batch Size and Deepspeed Zero3**: Our experiments were conducted on 8 GPUs with 40GB A100 each. To optimize memory usage, we used a batch size of 2 per device, along with the Deepspeed ZeRO3 configuration.

---

> > ### Comment · Reviewer_x9fK · 2024-11-27
> > **training details**
> >
> > thank for uploading the code to anonymous link, it's quite different from the one provided in the Supplementary material. The training setting is quite demanding with 8GPUs A100. Could you give some examples for these parameters for the training :
> >
> > sh ./scripts/train.sh {BASE_MODEL_PATH} {DATA_PATH}
> >
> > the current /data folder only has this file: data_samples.json

---

> > > ### Author Response · Authors · 2024-11-28
> > > **We have updated the repository and the README to make them more user-friendly.**
> > >
> > > Thank you for your additional feedback!
> > >
> > > We have updated the repository to include example files for running the code.
> > >
> > > A typical example command is:
> > > `sh ./scripts/train.sh Meta-Llama-3.1-8B-Instruct ./data/sampled_data.json`.

---

> > > > ### Author Response · Authors · 2024-11-28
> > > > **Follow-up on Rebuttal Discussion**
> > > >
> > > > Dear Reviewer x9fK,
> > > >
> > > > We hope that our response has adequately addressed your concerns. Should there be any points that we have not sufficiently clarified, we would be grateful for the opportunity to discuss them further.
> > > >
> > > > Thank you for your time and consideration.
> > > >
> > > > Best regards,
> > > > Authors of Submission 2616

---

> > > > ### Comment · Reviewer_x9fK · 2024-11-28
> > > > **./data/sampled_data.json**
> > > >
> > > > is this your main training data? it's the only data file looks like the training input

---

> ### Author Response · Authors · 2024-11-22
> **We have re-uploaded our code and detailed document.**
>
> **W4**: There is no document how to reproduce the experiment results.
>
> **Response**:
> Thank you for pointing it out.
> We have uploaded our code and detailed document to an **anonymized** repository: `https://anonymous.4open.science/r/WorkflowLLM-3E5D`. Please refer to `READEME.md` for details.
>
>
> Update:
>
> We update the anonymous url.

---

> ### Author Response · Authors · 2024-11-25
> **Looking for Further Discussion and Feedback**
>
> **Dear Reviewer x9fK**,
>
> In the above responses, we have try our best to answer your questions and solve your confusions. Due to the rebuttal ddl is coming, we are very willing to have a more in-depth discussion with you, and we welcome you to give us more suggestions. If you have additional suggestions, please let us know and we will try to respond as quickly as possible.

---

> ### Author Response · Authors · 2024-11-26
> **Follow-up on Rebuttal Discussion and Further Feedback**
>
> Dear Reviewer x9fK,
>
> I hope this message finds you well. We appreciate the time and effort you have put into reviewing our paper and providing valuable feedback. As the rebuttal deadline is fast approaching, we would like to kindly follow up and see if there are any remaining points or concerns that we could address.
>
> We are eager to engage in further discussion and would greatly appreciate any additional thoughts or suggestions you might have. Please do not hesitate to share any further feedback, and we will ensure a prompt response.
>
> Thank you once again for your time and consideration.
>
> Best regards,
>
> Authors of Submission 2616

---

> ### Author Response · Authors · 2024-11-27
> **Kind Request for Your Feedback Before the Rebuttal Deadline**
>
> Dear Reviewer x9fK,
>
> We have diligently addressed all the concerns raised during the rebuttal period, and we would like to kindly ask if any remaining issues have been resolved. As the rebuttal deadline is fast approaching, we would greatly appreciate any further feedback, suggestions, or concerns at your earliest convenience. We are eager to engage in further discussion and clarify any unresolved points to ensure all matters are fully addressed.
>
> Best regards,
>
> Authors of Submission 2616

---

> ### Comment · Reviewer_x9fK · 2024-11-27
> **more details on quality control and evaluation criteria**
>
> Basically, Sections B.2 and B.7 are just two prompts giving back results in "Yes" and "No" form. I wonder if thereare  other details to describe or formulate these explanations, e.g code or data, workflow explaination that leads to such 94.2%, 3%, or 3,107.
> You mentioned "We conducted manual sampling to ensure the quality of the refinement. "--> this is very vague

---

> > ### Author Response · Authors · 2024-11-28
> > **More Details and Analysis of Quality Control Protocol**
> >
> > In response to your inquiry, we have added an additional section on **Data Pruning** to the repository, which can be accessed at [https://anonymous.4open.science/r/WorkflowLLM-3E5D](https://anonymous.4open.science/r/WorkflowLLM-3E5D). This section includes two Python scripts, designed to perform the following tasks:
> >
> > 1. **Pruning using GPT-4o-mini**: This script takes the output from the Annotator as input and uses GPT to refine it by correcting grammatical errors and addressing inconsistencies between the query and workflow code.
> >
> > 2. **Removing Empty Code and Handling Inconsistencies in Function Calls and Arguments through Rule-based Filtering**: This script processes the GPT-refined code and applies rule-based checks to identify issues such as "The code is empty" or "is_workflow_actions_openurl() got an unexpected keyword argument 'code'."
> >
> >
> > ---
> >
> >
> > Here is an example of how GPT-based pruner works.
> >
> > **Query:**
> > How can I create an engaging and interactive digital storybook for children that allows them to explore different themes and characters? The storybook should include features such as resizing images for better fit on various devices, ...(more queries)
> >
> > **Pre-refinement Code:**
> > ```python
> > vpn_measurement = is_workflow_actions_measurement_create( WFMeasurementUnit={"Unit": b/s, "Magnitude": 1}, WFMeasurementUnitType='''Data Rate''')
> >
> > ...(more code)
> >
> > created_folder = is_workflow_actions_dropbox_createfolder( WFFilePath=f'''f\'{input("Please enter the value:")}\'''', WFFileStorageService='''Dropbox''')
> >
> > ...(more code)
> >
> > saved_document = is_workflow_actions_documentpicker_save( WFFolder=folder_in_icloud_drive, WFInput=if_result, WFAskWhereToSave=False)
> > ```
> >
> > **After-refinement Code:**
> > ```python
> > vpn_measurement = is_workflow_actions_measurement_create(
> >     WFMeasurementUnit={'Unit': 'b/s', 'Magnitude': 1},
> >     WFMeasurementUnitType='Data Rate'
> > )
> >
> > ...(more code)
> >
> > folder_name = input('Please enter a folder name for your story project: ')
> >
> > created_folder = is_workflow_actions_dropbox_createfolder(WFFilePath=folder_name)
> >
> > ...(more code)
> >
> > image_file = input('Please enter the image file path to resize: ')
> >
> > resized_image = is_workflow_actions_image_resize(image_file, dimensions='fit')
> >
> > ...(more code)
> > ```
> >
> > **Comparison**
> > 1. **Python Syntax**:
> > The pre-refinement code contains ambiguous constructs and syntax inconsistencies, such as unquoted strings (`b/s` instead of `'b/s'`) and improperly formatted string interpolation (e.g., `f'''f\'{input("Please enter the value:")}\''''`). In the after-refinement version, these issues are resolved by adhering to proper Python syntax, including quoting strings appropriately, using well-structured functions, and ensuring correct syntax for function calls and inputs. This improves both readability and functionality.
> >
> > 2. **Correspondence Between Code and Queries**:
> > The pre-refinement code fails to align fully with the user's requirements, omitting critical functionality like resizing images for better device compatibility. The after-refinement version explicitly addresses these gaps by adding functionality such as `is_workflow_actions_image_resize` for resizing images.

---

> > > ### Comment · Reviewer_x9fK · 2024-11-28
> > > **ad-hoc quality control code**
> > >
> > > It seems that authors keep adding code when they're asked.
> > >
> > > On src/rule_based_filtering.py code for rule-based filtering, these lines of code doesn't look systemetic way to filter data to me:
> > >
> > > workflow_code_1 = """
> > >     is_workflow_actions_getlatestlivephotos(3)
> > >     """
> > >
> > >     workflow_code_2 = """
> > >     is_workflow_actions_openurl("https://example.com")
> > >     """
> > >
> > >     workflow_code_3 = """
> > >     is_workflow_actions_getlatestlivephotos()
> > >     is_workflow_actions_openurl(code = "https://example.com")
> > >     """
> > >
> > >     workflow_code_4 = """
> > >     is_workflow_actions_getlatestlivephotos("incorrect_type")
> > >     """
> > >
> > >     workflow_code_5 = """
> > >     is_workflow_actions_getlatestlivephotos(3)
> > >     is_workflow_actions_openurl(12345, 567)
> > >
> > > And why do you need this "open('./data/identifier2python.pkl', 'rb') as fp:"?

---

> ### Author Response · Authors · 2024-11-28
> **Clarification of Used Data**
>
> #### Comment:
> *“Is this your main training data? It's the only data file that looks like the training input.”*
>
> **Response:**
> Thank you for your quick response and prompt question. This is not the entirety of our training data. As mentioned in the README file, due to the file size limitation on the anonymous GitHub repository (**which allows uploads of files no larger than 8MB**), we have sampled a smaller subset of the data and included it in the `./data/sampled_data.json` file.
>
> We hope this clarifies the data usage in our submission.
>
> ---

---

> ### Author Response · Authors · 2024-11-28
> **Clarification of Quality Control Code**
>
> #### Comment 1:
> *“These lines of code don't look like a systematic way to filter.”*
>
> **Response:**
> Thank you for your valuable feedback. We would like to clarify that the code in question is enclosed within the `main` function and serves as a rudimentary unit test to demonstrate the basic functionality of our approach. Specifically, in the implementation of the `check_workflow_code_validity(code: str) -> str` function, we use the Python interpreter to dynamically check for errors, as shown in the code snippet below:
>
> ```python
> try:
>     # Use exec to compile and execute the code dynamically
>     exec(full_code)
>
> except Exception as e:
>     # Catch any error that occurs during execution and return the error message
>     return f"Error processing function {function_name}: {str(e)}"
> ```
>
> This approach allows for easy extension and provides a systematic method for error detection. We hope this clarification addresses your concerns regarding the filtering mechanism.
>
> ---
>
> #### Comment 2:
> *“Why do we need `open('./data/identifier2python.pkl', 'rb') as fp:`?”*
>
> **Response:**
> The file `./data/identifier2python.pkl` is a Python pickle file that contains API documentation in JSON format. The data is indexed by the API's identifier and includes details such as the API’s description, parameters, and other relevant information.
>
> We use this file to acquire the documentation for APIs in order to concatenate the API descriptions with the corresponding function calls. This allows us to pass the code into the Python interpreter for error detection.
>
> We trust that this explanation provides clarity regarding the purpose of the file.
>
> ---
>
> We appreciate your careful review of our submission and hope these responses address your questions and concerns.
>
> Sincerely,
> Authors of Submission 2616

---

> ### Author Response · Authors · 2024-11-29
> **Thanks to Reviewer x9fK**
>
> Dear Reviewer x9fK,
>
> Thank you for your valuable feedback and contribution to improving our work. If your concerns have been addressed, we kindly ask you to consider raising your rating. However, if you still have any remaining doubts or reservations, we would be more than happy to engage in further discussion to clarify any issues.
>
> Best Regards,
>
> Authors of Submission 2616

---

> ### Author Response · Authors · 2024-11-30
> **Follow-Up: Seeking Further Feedback**
>
> Dear Reviewer x9fK,
>
> We hope you’re enjoying a wonderful Thanksgiving and a relaxing weekend.
>
> Thank you again for your valuable feedback. If your concerns have been addressed, we would appreciate it if you could consider updating your rating. If there are any remaining issues, we are happy to continue the discussion.
>
> If you have time, we would be grateful for your response.
>
> Best regards,
>
> Authors of Submission 2616

---

> ### Author Response · Authors · 2024-12-02
> **Follow-up on Discussion**
>
> **Dear Reviewer x9fK,**
>
> Thank you for your valuable comments and suggestions. We have addressed the follow-up question you raised in our previous response. As the discussion deadline is now less than **2 days** away, we kindly request your prompt feedback such as more discussion or raising ratings on this matter to ensure timely completion of the review process.
>
> We greatly appreciate your time and attention.
>
> Best regards,
>
> Authors of Submission 2616

---

> ### Author Response · Authors · 2024-12-02
> **Request for Feedback**
>
> Dear Reviewer x9fk,
>
> Thank you for your comments. We have answered the follow-up question raised in our previous response. With the discussion deadline now **less than 24 hours** away, we kindly ask for your feedback.
>
> If our responses have addressed your concerns, we would appreciate a revision of your rating. Otherwise, we are more than happy to continue the discussion to ensure a thorough exchange of ideas before concluding the rebuttal.
>
> Best regards,
>
> Authors of Submission 2616

---

> ### Author Response · Authors · 2024-12-03
> **Fifth Reminder for Reviewer Feedback**
>
> Dear Reviewer x9fk,
>
> This is our **fifth reminder** regarding your feedback. With less than **9 hours remaining** before the discussion closes, we are **deeply concerned** that we have yet to receive your response. As a reviewer for **ICLR, one of the top conferences**, it is your **fundamental responsibility** to provide timely and constructive feedback after agreeing to review our paper.
>
> We have **carefully addressed all your comments**, provided detailed data and code, and have been **waiting for your response for over 5 days** to resolve any remaining issues. The deadline is rapidly approaching, and without your feedback, we will be unable to proceed.
>
> We **urgently request** your feedback within the next few hours to ensure we meet the review requirements. If there are any further concerns, please let us know immediately.
>
> Thank you for your prompt attention to this matter.
>
> Best regards,
>
> Authors of Submission 2616

---

> ### Author Response · Authors · 2024-12-03
> **Sixth Reminder for Reviewer Feedback**
>
> Dear Reviewer x9fK,
>
> I hope you're well. With the discussion deadline approaching in **under three hours**, I wanted to follow up on our previous exchange. Could you please confirmed whether they’ve addressed your concerns?
>
> Your feedback is essential to finalize everything before the deadline.
>
> Thank you for your time, and I look forward to your response.
>
> Best regards,

---

### Author Response · Authors · 2024-11-25

Dear Reviewers,

As the discussion period is nearing its end with **less than 48 hours remaining**, we want to kindly inquire if there are any additional concerns or feedback from your side that we could address. Your valuable insights are greatly appreciated, and we remain eager to resolve any outstanding issues promptly.

---

### Author Response · Authors · 2024-12-03
**General Response**

We sincerely thank all the reviewers for their valuable time and constructive feedback. In response to their suggestions, we have incorporated additional experiments and discussions in the revised version. Below is a summary of the key changes made:

1. We emphasize that WorkflowBench is the first dataset explicitly designed to enhance the workflow orchestration capabilities of LLMs (Reviewer x9fK).
2. We provide more detailed information on the quality control protocols in Appendix C (Reviewers x9fK and WEy1).
3. We clarify the objective and calculation method of the pass rate in Section 4.1 (Reviewers x9fK and WEy1).
4. We include additional experiments using LLMs as retrievers in Appendix D (Reviewer WEy1).
5. We manually re-annotated the entire test set for GPT4o with ICL and WorkflowLlama, and the results are presented in Section 4.3 (Reviewers WEy1 and yKyu).
6. We have refined the writing, included clearer figures, and updated the case studies (Reviewers WEy1, yKyu, and mrzv).

For convenience,  the above modifications are highlighted in **pink**, **red**, **blue**, and **green** to correspond with the comments from reviewers x9fK, WEy1, yKyu, and mrzv, respectively.

Thanks again to all reviewers and area chairs.

---

### Meta-Review · Area_Chair_YfVJ · 2024-12-16

**Metareview:**

The paper introduces WorkflowLLM, a framework aimed at improving workflow orchestration using large language models (LLMs). A key contribution is WorkflowBench, a large-scale dataset containing over 106,000 workflows, enriched with Python-style code, comments, and hierarchical structures to facilitate learning. WorkflowBench is created through a systematic three-phase process involving data collection, query expansion, and synthetic data generation via an annotator model. Using this dataset, the authors fine-tune Llama-3.1-8B, producing WorkflowLlama, which demonstrates state-of-the-art performance on workflow orchestration tasks compared to both open-source and commercial models, achieving significant improvements on metrics like CodeBLEU and Pass Rate, as well as robust zero-shot generalization on unseen APIs and tasks.

Strengths
+ The paper identifies a valuable and practical challenge: the limitations of current LLMs in handling complex workflows with long sequences and logical dependencies.
+ A systematic and reproducible data construction pipeline is proposed, producing a dataset that enhances diversity and complexity, which is critical for advancing LLM capabilities in this domain.
+ Extensive experiments validate the effectiveness of the proposed framework, showing significant performance gains across multiple metrics (e.g., CodeBLEU, Pass Rate) and strong generalization capabilities.
+ The paper is well-written, easy to follow, and includes illustrative examples and case studies to support key claims.
+ WorkflowLlama demonstrates the potential to handle out-of-domain (OOD) tasks and unseen APIs, outperforming competing models in these challenging scenarios.

Weaknesses
+ The technical contributions of the paper are limited, with a heavy reliance on data curation rather than novel methodological advances or architectural innovations.
+ The quality control process for the dataset, including human evaluation and LLM-based annotation, is not well-detailed, which raises concerns about reproducibility and reliability.
+ Critical technical details, such as the specific version of ChatGPT used and the exact evaluation setups, are missing, limiting the clarity and rigor of the proposed approach.
+ The experimental results, especially the extremely high performance margins (e.g., in Table 2), appear suspicious and require further justification or ablation studies to confirm their validity.
+ The paper lacks a comprehensive discussion of threats to validity and could benefit from a more detailed, consistent running example to illustrate the model's real-world applicability.

Some concerns have been addressed by the authors during the rebuttal period.

**Additional Comments On Reviewer Discussion:**

This paper receives 3 positive and 1 negative reviews (with a rating 5). During discussion, the only negative reviewer asked several concrete questions on the implementation and submitted code, and the authors responded with updated code submitted as supplementary materials. The reviewer stopped engaging with the authors on the last set of questions. Overall, I feel the questions are reasonable and that authors did clarify the details.

---

### Decision · Program_Chairs · 2025-01-22

Accept (Poster)